# Progesterone induces meiosis through two obligate co-receptors with PLA2 activity

Nancy Nader[1,2], Lama Assaf[1,3], Lubna Zarif[1], Anna Halama[2,4], Sharan Yadav[1,5], Maya Dib[1], Nabeel Attarwala[6,7], Qiuying Chen[6], Karsten Suhre[2,4], Steven Gross[6], Khaled Machaca[1,2]*

[1]Calcium Signaling Group, Research Department, Weill Cornell Medicine Qatar, Education City, Qatar Foundation, Doha, Qatar; [2]Department of Physiology and Biophysics, Weill Cornell Medicine, New York, United States; [3]College of Health and Life Science, Hamad bin Khalifa University, Doha, Qatar; [4]Research Department, Weill Cornell Medicine Qatar, Education City, Qatar Foundation, Doha, Qatar; [5]Medical program, Weill Cornell Medicine Qatar, Education City, Qatar Foundation, Doha, Qatar; [6]Department of Pharmacology, Weill Cornell Medicine, New York, United States; [7]Biological Sciences division, University of Chicago, Chicago, United States

## eLife Assessment

This **important** study provides **solid** evidence for a non-genomic action of progesterone in *Xenopus* oocyte activation. The findings demonstrate that two non-genomic progesterone receptors, ABHD2 and mPRb, function as a novel progesterone-stimulated phospholipase A2. The findings will be of broad interest to reproductive endocrinologists and physiologists.

*For correspondence: khm2002@qatar-med.cornell.edu

Competing interest: The authors declare that no competing interests exist.

**Abstract** The steroid hormone progesterone (P4) regulates multiple aspects of reproductive and metabolic physiology. Classical P4 signaling operates through nuclear receptors that regulate transcription. In addition, P4 signals through membrane P4 receptors (mPRs) in a rapid nongenomic modality. Despite the established physiological importance of P4 nongenomic signaling, the details of its signal transduction cascade remain elusive. Here, using *Xenopus* oocyte maturation as a well-established physiological readout of nongenomic P4 signaling, we identify the lipid hydrolase ABHD2 (α/β hydrolase domain-containing protein 2) as an essential mPRβ co-receptor to trigger meiosis. We show using functional assays coupled to unbiased and targeted cell-based lipidomics that ABHD2 possesses a phospholipase A2 (PLA2) activity that requires mPRβ. This PLA2 activity bifurcates P4 signaling by inducing clathrin-dependent endocytosis of mPRβ, resulting in the production of lipid messengers that are G-protein coupled receptor agonists. Therefore, P4 drives meiosis by inducing an ABHD2 PLA2 activity that requires both mPRβ and ABHD2 as obligate co-receptors.

## Introduction

P4 signaling is critical for the regulation of female reproduction, sperm activation, neuronal function, and modulation of the immune system (*Garg et al., 2017*; *Thomas, 2012*; *Dressing et al., 2011*). The classical mode of P4 signaling is through nuclear receptors (nPRs) that act as transcription factors and modulate gene expression, resulting in cellular responses on a relatively slow time scale (*Rekawiecki et al., 2011*). In addition, P4 mediates rapid signal transduction that is independent of transcription

and has thus been referred to as nongenomic. This signaling modality can be mediated by mPRs that are members of the progestin and AdipoQ receptors family (PAQR), and consist of 11 receptors with 5 being specific to P4: PAQR5 (mPRγ), PAQR6 (mPRδ), PAQR7 (mPRα), PAQR8 (mPRβ), and PAQR9 (mPRϵ) (*Tang et al., 2005*). Several lines of evidence support the importance of nongenomic mPR-dependent signaling, including results from studies of nPR knockout mouse lines, the speed of the signal transmission following P4 treatments, and the observed activity of membrane impermeant BSA-coupled P4 (*Dressing et al., 2011*; *Valadez-Cosmes et al., 2016*; *Frye et al., 2006*; *Lydon et al., 1995*). Currently, mPR-mediated signaling is recognized as an important regulator of many biological functions in the nervous and cardiovascular systems, female and male reproductive tissues, intestine, immune and cancer cells, and glucose homeostasis (*Dressing et al., 2011*; *Valadez-Cosmes et al., 2016*; *Moussatche and Lyons, 2012*; *Dosiou et al., 2008*; *Flock et al., 2013*). Thus, mPRs are emerging as potential clinical targets for hypertension, reproductive disorders, and neurological diseases (*Wendler and Wehling, 2022*). However, their downstream signaling events are not well defined.

mPRs are integral membrane proteins with predicted 7-transmembrane domains, a cytoplasmic N-terminus, and an extracellular C-terminus, which were first identified in fish ovaries almost two decades ago (*Tang et al., 2005*; *Zhu et al., 2003*; *Nader et al., 2020*). This topology is opposite to that of G-protein coupled receptors (GPCRs), yet there is an abundance of evidence for G protein activation downstream of mPRs, primarily $G\alpha_i$ and also $G\beta\gamma$ (*Thomas, 2022*). However, there are conflicting reports as to whether mPRs signal through trimeric G-proteins or other modalities that remain to be defined (*Moussatche and Lyons, 2012*; *Smith et al., 2008*).

The anuran *Xenopus laevis* is a particularly suitable model to study P4 nongenomic signaling for two reasons: (1) the oocyte is transcriptionally silent, so signaling can only occur through the nongenomic arm; (2) P4 is known to trigger *Xenopus* oocyte maturation through the activation of mPRβ (PAQR8) (*Nader et al., 2020*; *Josefsberg Ben-Yehoshua et al., 2007*). Fully grown vertebrate oocytes arrest at prophase I of meiosis for prolonged periods before undergoing maturation in preparation for fertilization (*Nader et al., 2013*). Importantly, P4 triggers re-entry into meiosis as well as a series of cytoplasmic differentiation steps that allow the egg to become fertilization-competent, able to activate in response to sperm entry, and to support early embryogenesis (*Machaca, 2007*). Signaling downstream of mPRβ ultimately leads to activation of maturation promoting factor (MPF), composed of Cdk1 and cyclin B, that serves as the primary kinase which triggers oocyte entry into M-phase (*Nebreda and Ferby, 2000*). MPF is activated non-reversibly through two signaling cascades, Plk-Cdc25C and the MAPK cascade (*Figure 1A*).

In addition, oocyte meiotic arrest is maintained by high levels of cAMP-PKA through the action of a constitutively active GPCR, GPR185 (*Ríos-Cardona et al., 2008*; *Nader et al., 2014*). PKA blocks maturation by inhibiting both arms of the pathway: translation and Cdc25C (*Figure 1A*; *Duckworth et al., 2002*; *Matten et al., 1994*). Interestingly, GPR185 is internalized in response to P4, which inhibits its ability to activate adenylate cyclase and thus renders the oocyte more permissive for maturation (*Nader et al., 2014*). Several studies have detected a drop in cAMP levels in response to P4 using ELISA (*Nader et al., 2014*; *Schorderet-Slatkine et al., 1982*; *Maller et al., 1979*; *Sadler and Maller, 1981*; *Nader et al., 2016*), but these findings could not be replicated using various reporters for cAMP and PKA at the single-cell level in real-time (*Nader et al., 2016*). Furthermore, the different sensors did not detect a global drop in cAMP in response to P4, and P4-dependent maturation proceeds even when cAMP is high (*Nader et al., 2016*). This argues that P4 signals through parallel pathways that can overwhelm the cAMP-PKA-dependent inhibition.

We have previously shown that mPRβ signals through APPL1 and AKT2 to activate Plk and MPF by forming a complex of activators within signaling endosomes (*Nader et al., 2020*). Importantly, enrichment of mPRβ within endosomes is sufficient to induce maturation in the absence of P4, arguing that a primary function of P4 is to induce clathrin-dependent endocytosis of mPRβ (*Nader et al., 2020*).

The lipase α/β hydrolase domain-containing protein 2 (ABHD2), an integral plasma membrane protein belonging to the ABHD family, was identified as a membrane progesterone receptor in human sperm (*Wendler and Wehling, 2022*; *Miller et al., 2016*) ABHD2 acts a monoacyl glycerol lipase (MAGL) that hydrolyses 2-arachidonoylglycerol (2-AG) forming arachidonic acid (AA) and glycerol and leads to sperm activation (*Miller et al., 2016*). ABHD2 has also been implicated in follicular maturation in mammals (*Björkgren et al., 2021*), vascular smooth muscle migration, and in pulmonary

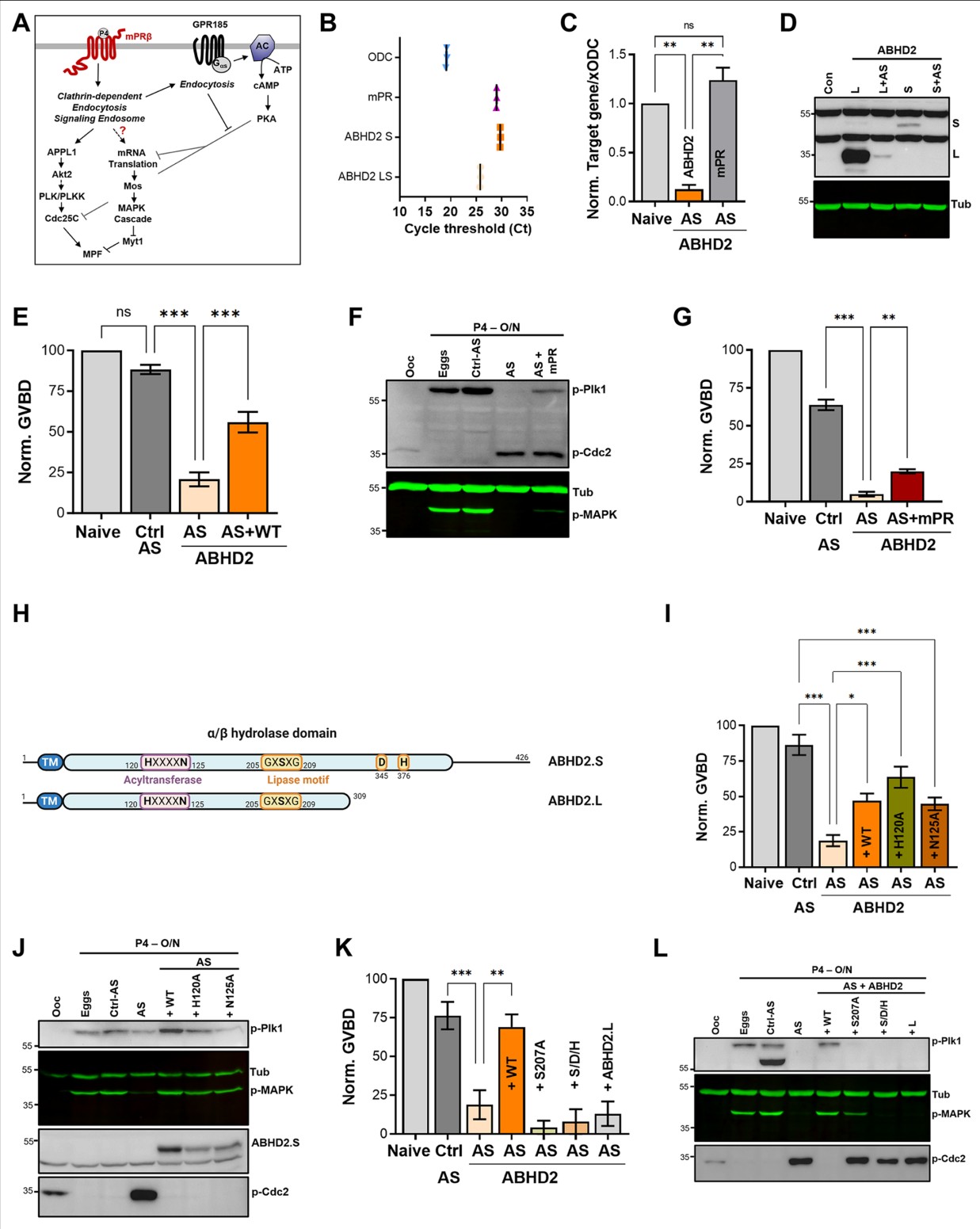

**Figure 1.** α/β hydrolase domain-containing protein 2 (ABHD2) is required for P4-induced oocyte maturation. (**A**) Signaling events downstream of mPRβ after progesterone (P4), leading to oocyte maturation. The role of GPR185 is also depicted. (**B**) mRNAs transcripts levels of mPRβ, ABHD2.S, ABHD2.LS, and *Xenopus* Ornithine decarboxylase (xODC) in oocytes using the Cycle threshold (Ct) generated from real-time PCR. (**C**) ABHD2 knockdown experiments. Oocytes were injected with specific ABHD2 antisense (AS) oligonucleotides and incubated at 18 °C for 24 hr. RNAs were prepared and analyzed by RT-PCR to determine the efficacy of ABHD2 knockdown as compared to naïve oocytes and mPRβ RNAs levels. Data are expressed as relative RNAs levels of ABHD2 and mPRβ mRNA after normalizing to xODC as a house keeping gene. (**D**) Representative WB of

*Figure 1 continued on next page*

*Figure 1 continued*

ABHD2 in naive and oocytes over-expressing ABHD2.L or ABHD2.S, following ABHD2 antisense (AS) oligonucleotide injection. Tubulin is shown as a loading control. (**E**) Oocyte maturation following injection of control antisense (Ctrl AS) or ABHD2 antisense (AS) with or without overexpression of ABHD2.S (AS +WT) and normalized to P4-treated naïve oocytes condition (Naive), (mean ± SEM; n=7 independent female frogs, ordinary one-way ANOVA). (**F**) Representative WB of MAPK, Plk1, and Cdc2 phosphorylation from untreated oocytes, eggs matured by overnight (O/N) treatment with P4, oocytes injected with control antisense (Ctrl AS) or ABHD2 antisense (AS) with or without overexpression of mPRβ. Tubulin is shown as a loading control. (**G**) Oocyte maturation in oocytes injected with control antisense (Ctrl AS) or ABHD2 antisense (AS) with or without overexpression of mPRβ and normalized to P4-treated naïve oocytes (mean ± SEM; n=3 independent female frogs, ordinary one-way ANOVA). (**H**) Schematic representation of *Xenopus* ABHD2.S and.L domains, including the acyltransferase and lipase motifs. (**I, K**) Oocyte maturation in oocytes injected with control antisense (Ctrl AS) or ABHD2 antisense (AS) with or without overexpression of ABHD2.S wild type (AS +WT) and the different ABHD2.S mutants as indicated, and normalized to P4-treated naïve oocytes (mean ± SEM; n=3 independent female frogs, ordinary one-way ANOVA). (**J,L**) Representative WB of MAPK, Plk1, and Cdc2 phosphorylation from untreated oocytes, eggs matured by overnight (O/N) with P4, oocytes injected with control antisense (Ctrl AS) or ABHD2 antisense (AS) with or without overexpression of the different ABHD2.S mutants as indicated on the panel. Tubulin is shown as a loading control.

The online version of this article includes the following source data and figure supplement(s) for figure 1:

**Source data 1.** Original files for western blot analysis displayed in *Figure 1D,F,J,L*.

**Source data 2.** File containing labeled western blots for *Figure 1D,F,J,L*, indicating the relevant bands and treatments.

**Figure supplement 1.** Progesterone receptors in Xenopus oocytes.

**Figure supplement 1—source data 1.** Original files for western blot analysis are displayed in *Figure 1C,D*.

**Figure supplement 1—source data 2.** File containing labeled western blots for *Figure 1C,D*, indicating the relevant bands and treatments.

emphysema (*Jin et al., 2009*; *Miyata et al., 2005*). In addition, ABHD2 was proposed to function as a triacylglycerol (TAG) lipase (*Naresh Kumar et al., 2016*), and was shown to localize to the ER where it regulates $Ca^{2+}$ release (*Yun et al., 2017*).

Here, we show that ABHD2 is required for the P4-induced release of oocyte meiotic arrest. Using untargeted lipidomic analyses, we detect broad downregulation of glycerophospholipid (GPL) and sphingolipid lipid metabolites in association with the enrichment of a key bioactive lipid messengers that include prostaglandins (PGs), lysophosphatidic acid (LPA), and potentially sphingosine-1-phosphate (S1P). Importantly, we show that ABHD2 acts as a PLA2 that requires mPRβ to generate the aforementioned lipid messengers. The ABHD2 evoked PLA2 activity also triggers mPRβ endocytosis, which as we have previously shown is sufficient to initiate oocyte maturation (*Nader et al., 2020*). To our knowledge, this is the first example of an α/β hydrolase that requires a heterologous co-receptor for activation.

## Results

### ABHD2 is required for P4-induced oocyte maturation

While reviewing the expression levels of various progesterone receptors in the *Xenopus* ovary, we noticed high levels of expression of ABHD2 in the oocyte and egg (*Figure 1—figure supplement 1A*). *Xenopus laevis* is tetraploid so it expresses two different versions of each gene, called S and L. There is significant sequence conservation between xABDH2.S and xABDH2.L with the human isoform, expect that xABDH2.L is truncated at position 309 (*Figure 1—figure supplement 1B*). Using primers specific to xABDH2.S and ones that amplify both the S and L isoforms, we confirmed that ABHD2 RNA is expressed in the oocyte at levels similar to those of mPRβ RNA (*Figure 1B*).

To test whether ABHD2 plays a role in oocyte maturation, we downregulated its expression by injecting anti-ABHD (L/S) oligos, which resulted in a significant reduction of ABHD2 RNA levels without affecting the levels of mPRβ RNA (*Figure 1C*). We could not detect endogenous oocyte ABHD2 using Western blots (WB) most likely due to low expression levels, as we could readily detect overexpressed ABHD2 (*Figure 1D*). We thus tested the efficacy of the antisense (AS) approach on overexpressed ABHD2 (either L or S). *Figure 1D* shows the effective reduction of both isoforms (*Figure 1D*) within 24 hr after antisense injection, arguing for a short ABHD2 half-life on the order of hours. Interestingly, knocking down ABHD2 expression inhibited oocyte maturation in response to P4 (*Figure 1E*), and was coupled to blocking Plk1, MAPK, and MPF activation (*Figure 1F*). This inhibition was significantly reversed by overexpression of ABHD2 (*Figure 1E*), confirming that the specific knockdown of ABHD2 – and not off-target effects – mediated AS action. Overexpression of mPRβ in ABHD2 antisense treated oocytes (*Figure 1—figure supplement 1C*) was less effective at rescuing

oocyte maturation although it did have a significant effect (*Figure 1G*), and partially rescued Plk1 and MAPK activation but not MPF (*Figure 1F*). These data argue that both mPRβ and ABHD2 are required to sufficiently activate the kinase cascades downstream of P4 to induce oocyte maturation.

## ABHD2 lipase domain is required for P4-induced oocyte maturation

ABHD2 is a serine hydrolase that belongs to the α/β hydrolase family with a conserved lipid hydrolase domain GxSxG with residues D345 and H376 being important for lipase activity based on sequence homology (*Figure 1H*). In addition, as with other members of the α/β hydrolase family, *Xenopus* ABHD2 has an acyltransferase domain that is close to the human HxxxD consensus but with Asn replacing the Asp residue (*Figure 1H*). As ABHD2 has been shown to function as a P4-dependent hydrolase that is important for sperm activation (*Miller et al., 2016*), we wondered whether ABHD2 lipase activity is similarly important for oocyte maturation.

To evaluate this possibility, we replaced the conserved functionally relevant residues in the HxxxD acyltransferase domain motif (i.e. H120 and N125) with Ala and tested the mutants' ability to induce oocyte maturation in response to P4. Both the H120A and N125A mutants induced oocyte maturation to similar levels as those observed with WT ABHD2 in oocytes injected with ABHD2-AS (*Figure 1I*), and this rescue was associated with effective expression of both mutants and activation of Plk1, MAPK, and MPF in eggs (*Figure 1J*). These findings demonstrate that the ABHD2 acyltransferase motif is not required for maturation.

To test for the role of ABHD2 lipid hydrolase activity in oocyte maturation, we mutated all three catalytically relevant residues: S207 within the GxSxG motif, D345, and H376 to Ala (S/D/H mutant) or just S207 to Ala. Effective expression of either the S/D/H or S207A mutants (*Figure 1—figure supplement 1D*), did not induce maturation in oocytes where endogenous ABHD2 was knocked down (*Figure 1K*). Neither mutant activated Plk1 or MPF, but expression of S207A led to low-level MAPK activation (*Figure 1L*) that was not associated with maturation (*Figure 1K*). We further tested whether the shorter ABHD2.L (309 residues) is functional as it is missing the C-terminal end of the α/β hydrolase domain including both D345 and H376. Notably, ABHD2.L did not induce oocyte maturation in response to P4 (*Figure 1K*) nor did it activate Plk1, MAPK, or MPF (*Figure 1L*). These data show that all three residues involved in lipase activity (S207, D345, H376) are critical for ABHD2 P4-mediated oocyte maturation.

## Lipidomics during oocyte maturation

The requirement for the lipase domain within ABHD2 for P4-dependent oocyte maturation suggests a potential role for lipid messengers downstream of P4 to release oocyte meiotic arrest. There is support for this idea in the literature from several early studies implicating lipid messengers in oocyte maturation, although there has been little consensus regarding specific pathways, lipases, or lipid mediators (*Mostafa et al., 2022*). To globally assess oocyte lipid profiles in response to P4, we performed unbiased mass-spectrometry-based lipidomics and metabolomics (using the Metabolon CLP and HD4 platforms) at two-time points after P4: 5 min to assess rapid changes in lipid abundances, and 30 min, a 'point of no return' where the majority of oocytes in the population commit to maturation (*Nader et al., 2014*). Furthermore, to be able to differentiate between pathways that are activated specifically through mPRβ or ABHD2, we performed the profiling on oocytes following mPRβ or ABHD2 knockdown.

The role of lipid signaling in oocyte maturation remains poorly defined, partly due to technical limitations in early studies, but also as a result of the complexity and transient nature of lipid signals (*Mostafa et al., 2022*). An additional confounding factor in oocyte lipidomics is that oocytes in the population mature in an asynchronous fashion in response to P4. This asynchrony makes the identification of transient lipid messengers challenging. Finally, metabolomic profiles depend on environmental factors, basal metabolic rates, diet, and physical activity and are as such quite variable. We observe this variability as metabotypes that are specific to individual frog donors in the principal component analysis in data from both the HD4 and CLP platforms (*Figure 2—figure supplement 1A*).

To minimize these technical and biological confounding factors, we performed the metabolomics analysis on groups of 10 oocytes at each time point and looked for relative changes in specific metabolites over time for each donor (*Figure 2A*). That is the metabolic profile at T0 served as the baseline for each frog and we looked for changes in each metabolite at the 5- and 30 min time points after P4

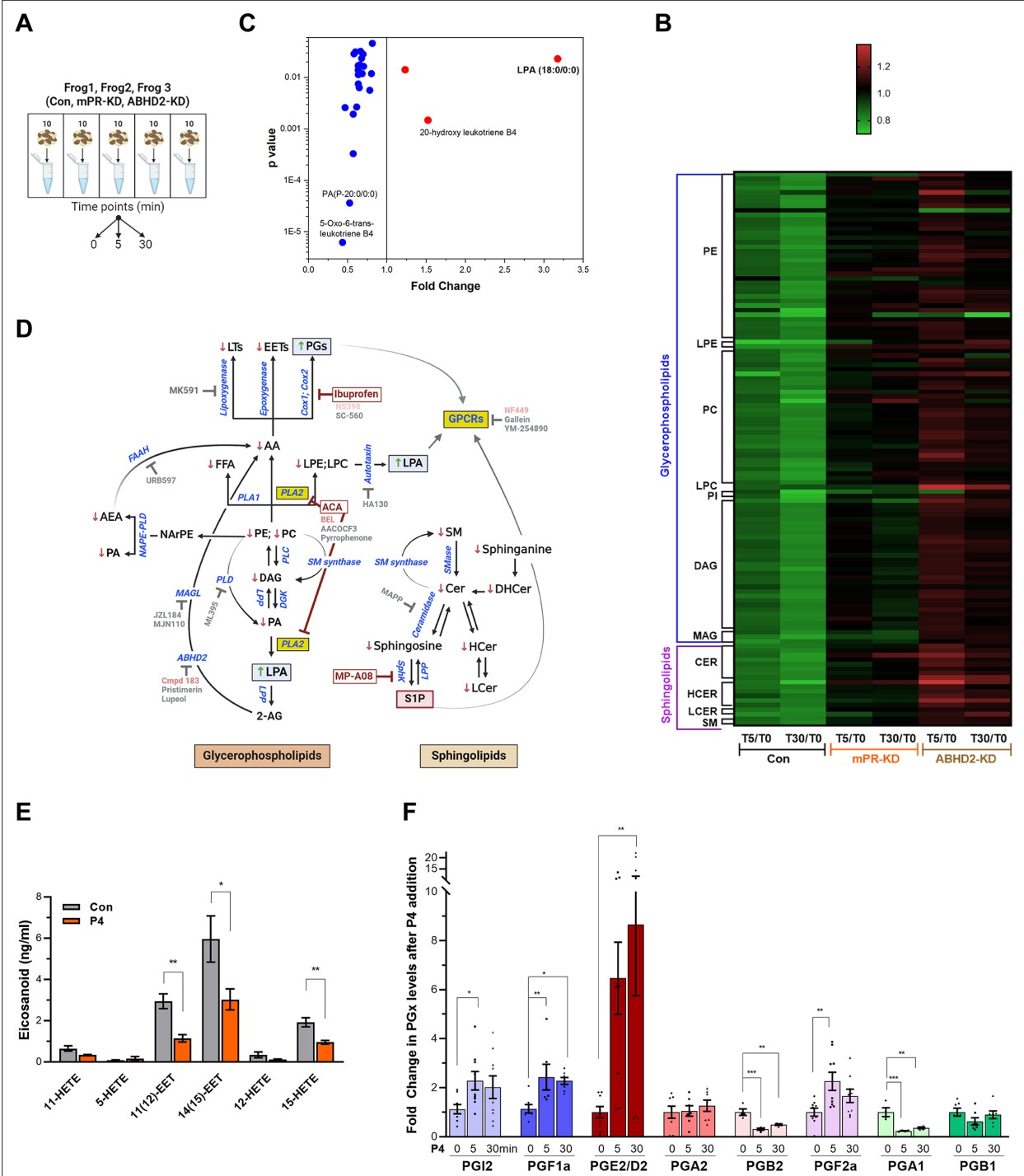

**Figure 2.** Metabolomics and lipidomics. (**A**) Summary of the experimental setup. Naïve oocytes (Con) and oocytes injected with either mPR (mPR-KD) or α/β hydrolase domain-containing protein 2 (ABHD2) antisense (ABHD2-KD) were treated for 5 or 30 min with progesterone (P4). For each condition and time point, five replicates of 10 pooled oocytes each were collected. The experiment was repeated using three donor females. (**B**) Heatmap of fold changes for metabolites that were changed significantly (p<0.05) at either the 5 min (T5) or 30 min (T30) time points in response to P4 as compared to untreated oocytes (T5/T0 and T30/T0) for naive (Con), mPR-KD, and ABHD2-KD oocytes. Metabolites are clustered at the levels of glycerophospholipids and sphingolipids and then at the pathway level as follows: PE (Phosphatidylethanolamines), LPE (Lysophosphatidylethanolamines), PC (Phosphatidylcholines), LPC (Lysophosphatidylcholine), PI (Phosphatidylinositols), DAG (Diacylglycerols), MAG (Monoacylglycerols), CER (Ceramides), HCER (Hexosylceramides), LCER (Lactosylceramides), and SM (Sphingomyelins). *Source data 3* Tables 1–6 show the means and p-value for each ratio fold change. (**C**) Distribution of metabolites that are reduced (blue) or increased (red) significantly in single naïve oocytes 30 min after P4. Fold change and p-values were calculated from 10 individual oocytes at each time point. The raw data is listed in *Source data 3* Table 7. (**D**) Summary of the changes

*Figure 2 continued on next page*

*Figure 2 continued*

in sphingolipids and glycerophospholipids after progesterone treatment. Increase (upward green arrow) or decrease (downward red arrow) in metabolite levels is noted. Tested chemical inhibitors are also shown. Strong inhibitors are indicated in red, weak inhibitors in pink, and drugs that do not inhibit oocyte maturation in gray. Generated using Biorender. (**E**) Levels of EETs and HETEs before and after P4 treatment in single oocytes. Naïve oocytes were incubated with either ethanol or P4 $10^{-5}$M for 30 min. 20 single oocytes per condition were collected and used for analysis. 5-oxoETE and 8 (9)-EET were detected in 1 or 2 samples, respectively, so they were not included in the statistical analyses although both were lower following P4 treatment. (**F**) Levels of prostaglandins before (0 min) and after P4 treatment at 5 min and 30 min time points in naïve oocytes. Per each condition, 10 replicates were collected containing 10 pooled oocytes each. The following eicosanoids were not detected in either group: 6kPGF1α, PGF2α, PGE2, TXB2, PGD2, PGA2, PGJ2, 15-deoxyPGJ2, 12-HHTrE, 11-dehydro TXB2, LTB4, LTC4, LTD4, LTE4, 20-hydroxy LTB4, 20-carboxy LTB4, 5 (6)-DiHETEs, LXA4, LXB4, 5 (6)-EET, 5 (6)-DiHET, 8 (9)-DiHET, 11 (*Wendler and Wehling, 2022*)-DiHET, 14 (*Thomas, 2022*)-DHET, 20-HETE. Similar results were obtained from individual oocyte samples (see *Source data 3* Table 7). Data are normalized to the PG levels at time zero. Example of the raw abundance of individual PG species is shown in *Figure 2—figure supplement 1C*.

The online version of this article includes the following figure supplement(s) for figure 2:

**Figure supplement 1.** Metablomics profiles during oocyte maturation.

addition. Furthermore, we averaged data from specific metabolites at the sub-pathway level to identify changes that are consistent among the three donor frogs tested. Data at the specific metabolite levels for both the CLP and HD4 platforms are presented in *Source data 3*.

Lipidomics analyses on the CLP platform show downregulation in response to P4 of multiple sphingolipid species, including ceramide, hexosylceramide, lactosylceramide, and sphingomyelin at both the 5 min and 30 min time points (*Figure 2B* and *Source data 3* Tables 1–6). A similar decrease in multiple glycerophospholipid species, including phosphatidylethanolamine (PE), phosphatidylcholine (PC), lysoPE, lysoPC, as well as glycerides including monoacylglycerol (MAG), and diacylglycerol (DAG) is observed (*Figure 2B* and *Source data 3* Tables 1–6). The HD4 platform analysis supports and extends these findings although at a lower level of granularity in terms of specific lipid species showing downregulation of multiple fatty acid species including AA (*Figure 2—figure supplement 1B*). Note that most of these changes at the specific biochemical levels are relatively small (20–25% decrease) yet they gain relevance through their combined trend through multiple experiments at the levels of multiple metabolites within the same pathway (*Source data 3* Tables 1–6).

Importantly, knockdown of either ABHD2 or mPRβ eliminated the downregulation of sphingolipids and glycerophospholipids in response to P4 at both the 5- and 30 min time points (*Figure 2B* and *Figure 2—figure supplement 1B*). As oocytes with mPRβ or ABHD2 knockdown do not mature in response to P4, the lipidomics data argue that the changes observed in the lipid metabolites are important for oocyte maturation; and that both mPRβ and ABHD2 are required to initiate the lipase activities underlying these changes. The fact that many of those changes are observed early on after P4 addition (5 min) argues that they represent the initial signaling step downstream of P4 to commit the oocyte to meiosis.

We were initially puzzled by the global lipidomics profiles as they showed mostly downregulation of both GPL and sphingolipid species. If the cells are indeed in broad lipid catabolism mode, one would expect enrichment in some lipid end products. However, because oocyte maturation is asynchronous, changes in individual lipid species may be masked when several oocytes are pooled as in our lipidomics analyses. To get better insights into changes at the single oocyte level, we developed approaches (see Methods) to extract and analyze metabolites at the single oocyte level at the 30 min post-P4 time point. Single oocyte data showed a threefold increase in LPA and replicated the downregulation of GPL and sphingolipid species observed in the pooled analysis (*Figure 2C*). The lipidomics changes observed in response to P4 are summarized in *Figure 2D* along both the GPL and sphingolipid pathways by an arrow up (red) or down (green) next to each detected lipid species.

Some end products of GPL and SL metabolism such as eicosanoids and sphingosine 1 phosphate (S1P) are not well suited to the standard extraction protocols used for our global metabolomics studies. Therefore, to enrich for these end products we performed targeted extraction and MS analyses against validated standards for eicosanoids and S1P, which would be the most likely end metabolites enriched based on the downregulation profile of other metabolites throughout the GPL and SL pathways (*Figure 2D*). AA is a precursor for eicosanoids, which are produced through three main enzymatic pathways: (1) cyclooxygenases (cox1/cox2) action results in the formation of prostaglandins; (2) lipoxygenases produce leukotrienes; and (3) cytochrome P450 enzymes, $\omega$-hydrolases

and epoxygenases, yield hydroxyeicosatetraenoic acids (HETEs) and epoxyeicosatrienoic acids (EETs), respectively (*Figure 2D*).

We could not detect any leukotrienes in the oocyte samples despite testing for multiple species (LTB4, LTC4, LTD4, LTE4, 20-hydroxy LTB4, 20-carboxy LTB4) but could observe a downregulation in their precursor hydroxy-eicosatetraenoic acids (HETE) (*Figure 2E*). We did, however, detect a decrease in LTB4 derivatives in the single oocyte MS analyses (*Figure 2B*). We also observed decreases in two epoxyeicosatrienoids (EET) from single oocyte samples (20 oocytes at 30 min post-P4) (*Figure 2E*). In contrast to leukotrienes and lipoxins, the trend in prostaglandins (PG) was the opposite with increases observed in multiple species, including PGI2, PGF1a, PGE2/PGD2, and PGF2a (*Figure 2F*). *Figure 2E* shows the relative changes of PG species from different donor females. To illustrate the basal abundance of the different PGs species relative to each other, *Figure 2—figure supplement 1C* shows the raw abundance of the different PGs relative to each other. Collectively these targeted metabolomics profiles show that AA is preferentially metabolized by cyclooxygenases at the expense of lipoxygenases/epoxygenases to produce PGs in response to P4 (*Figure 2D*).

From the metabolomics analyses of the SL pathway, sphingosine-1-phosphate (S1P) is predicted to increase in response to P4, as all other SL metabolites decrease in response to P4 (*Figure 2D*). We, therefore, directly tested S1P levels in the oocyte in response to P4 at 5 and 30 min but did not detect any significant change (*Figure 2—figure supplement 1D*). This implies that either S1P does not change in response to P4, or alternatively that the S1P produced in response to P4 diffuses out of the oocyte. This is possible as S1P is membrane-permeant and acts on cell surface receptors from the extracellular side (*Brindley and Pilquil, 2009*). We tested this possibility directly below by assessing the role of S1P receptors in oocyte maturation.

## Pharmacological validation of the metabolomics findings

The metabolomics findings from extensive analyses using multiple platforms and extraction procedures to enrich and detect specific lipid species argue for an important role for lipid messengers in inducing oocyte maturation. This is further supported by the fact that when either mPRβ or ABHD2 are knocked down the lipid changes disappear. However, these metabolomics studies are correlative and do not address cause-effect relationships. To test whether activation of specific lipases is required for oocyte maturation, we used pharmacological tools to inhibit specific enzymes predicted to be important for maturation based on our metabolomic analyses (*Figure 2D*).

Along the GPL metabolic arm, the lipidomics data show the downregulation of upstream metabolites with the enrichment of LPA and prostaglandins (PGs) (*Figure 2*). This argues for a central role in PLA2 activity as it would produce AA as a precursor for PGs and LPC/LPE as precursors for LPA. To test for a role for PLA2 in maturation, we used a broad-spectrum inhibitor, ACA, as PLA2s represent a large family of enzymes (up to 16 groups) with disparate activation modes, $Ca^{2+}$-dependency, and subcellular localization (secreted or cytosolic) (*Dennis et al., 2011*; *Leslie, 2015*). ACA completely inhibited maturation with an $IC_{50}$ of $3.8 \times 10^{-6}$ M (*Figure 3A*) in line with its reported PLA2 $IC_{50}$ value ($5 \times 10^{-6}$ M, see *Table 1*). ACA treatment blocked Plk1, MAPK, and MPF activation explaining the maturation inhibition (*Figure 3B*). Given the robust inhibition with ACA, we tested other PLA2 inhibitors, including AACOF3, bromoenol lactone (BEL), and pyrrophenone to get insights into the PLA2 isoform involved. BEL inhibited oocyte maturation with an $IC_{50}$ of $3.9 \times 10^{-5}$ M (*Figure 3A*), by mainly blocking the Plk1 pathway (*Figure 3C*). The BEL $IC_{50}$ was about fivefold higher than its reported inhibitory potency of $8 \times 10^{-6}$ M (*Table 1*). The other two PLA2 inhibitors were ineffective (*Figure 3A*). It is difficult to pinpoint a particular PLA2 isoform based on the isoform-specific profile of these inhibitor's (see https://www.sigmaaldrich.com/QA/en/technical-documents/technical-article/protein-biology/protein-expression/phospholipase-a2), arguing that the oocyte's PLA2 activity required for maturation does not match known PLA2 isoforms. Nonetheless, the inhibitors data support a central role for PLA2 in inducing oocyte maturation downstream of P4. In line with this, pre-treatment of oocytes with mastoparan, a peptide toxin from wasp venom known to activate PLA2, enhanced oocyte maturation by ~50% in response to low P4 (*Figure 3D*).

AA could still be produced through the action of fatty acid amide hydrolase (FAAH) independently of PLA2 activation, and PLD activation produces phosphatidic acid (PA) which is a precursor of LPA (*Figure 2F*). Inhibition of either FAAH or PLD had no effect on oocyte maturation (*Figure 3—figure*

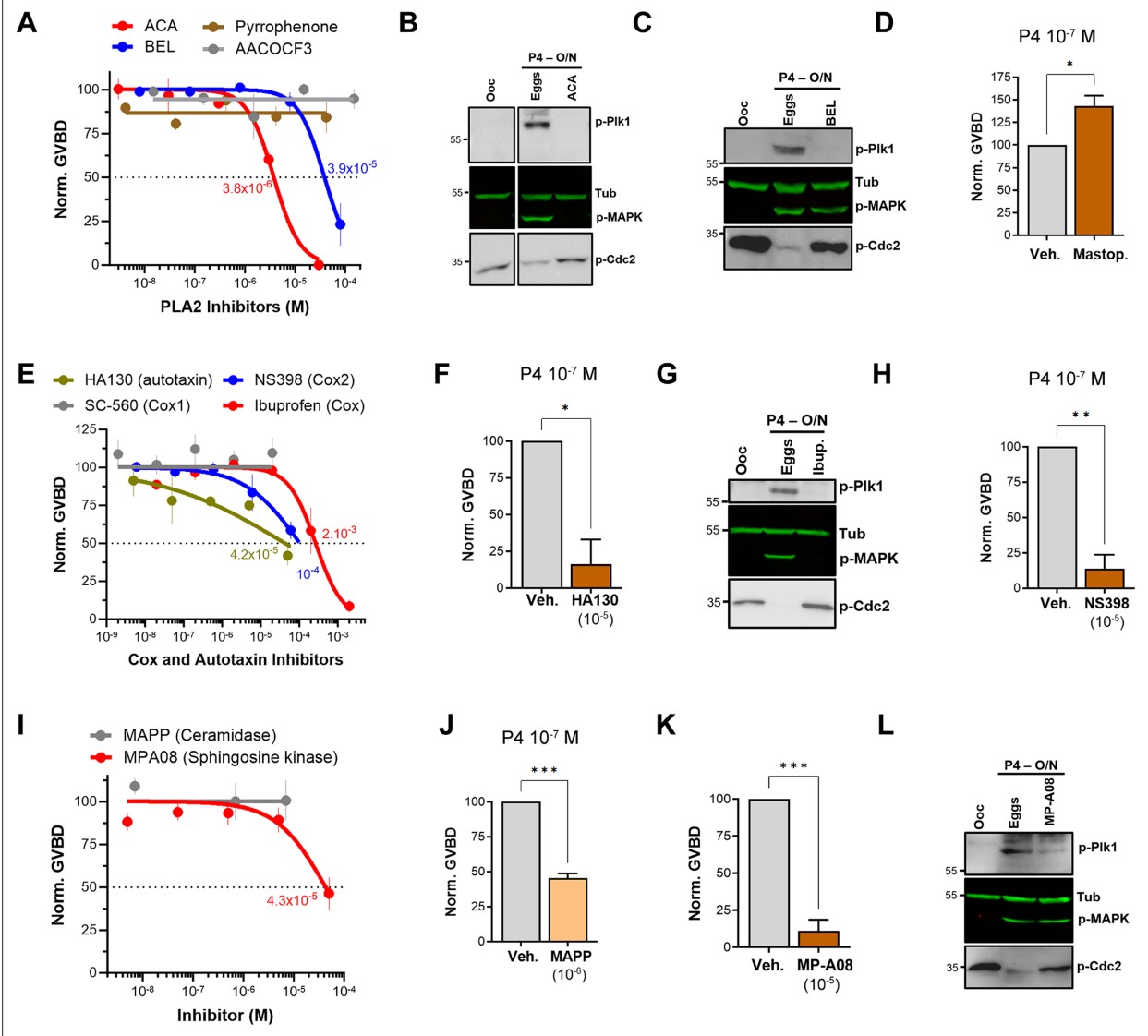

**Figure 3.** Pharmacological validation of the metabolomics findings. (**A, E, I**) Dose-response of inhibition of oocyte maturation for the drugs tested. Oocytes were pre-treated for 2 hr with a vehicle or with increasing concentrations of the indicated drugs, followed by overnight treatment with progesterone (P4) at $10^{-5}$ M. IC$_{50}$ was calculated using a nonlinear regression fit (mean ± SEM; n=3 independent female frogs). (**D, F, H, J, K**) Drug effect on oocyte maturation at low P4 concentration. Oocytes were pre-treated for 2 hr with the vehicle or with the highest drug concentration from the dose-response, followed by overnight treatment with P4 at $10^{-7}$ M. Oocyte maturation was normalized to control oocytes (treated with vehicle) (mean ± SEM; n=3 independent female frogs for each chemical compound experiment, unpaired t-test). (**B, C, G, L**) Representative WB of MAPK, Plk1, and Cdc2 phosphorylation from untreated oocytes, oocytes pretreated with vehicle or the indicated drug for 2 hr then matured overnight (O/N) with P4 (eggs). Tubulin is shown as a loading control.

The online version of this article includes the following source data and figure supplement(s) for figure 3:

**Source data 1.** Original files for western blot analysis are displayed in *Figure 3B,C,G,L*.

**Source data 2.** File containing labeled western blots for *Figure 3B,C,G,L*, indicating the relevant bands and treatments.

**Figure supplement 1.** Effects of various pharmacological inhibitors on oocyte maturation.

supplement 1A), arguing that the precursors for LPA and PGs are produced primarily through PLA2 activation in response to P4 in the oocyte.

Given the inhibitory effect of ABHD2 knockdown on oocyte maturation, we tested the effect of two plant triterpenoids known to inhibit ABHD2 and MAGLs, pristimerin and lupeol (*Mannowetz et al., 2017*), as well as MAGL chemical inhibitors JZL184 and MJN110. None of these inhibitors had a significant inhibitory effect on oocyte maturation at high P4 (*Figure 3—figure supplement 1B*). However,

**Table 1.** Pharmacological inhibitors potency.

| Inhibitor | Target | IC$_{50}$ ± SEM Oocyte maturation (M) | IC$_{50}$ (M) Representative study | Ratio (Ooc/Lit) |
|---|---|---|---|---|
| ACA | PLA2 | 3.8E-6±5.8E-7 | 5E-6 *Liu, 1999* | 0.76 |
| BEL | PLA2 | 3.9E-5±8E-6 | 8E-6 *Balsinde and Dennis, 1996* | 4.9 |
| MP-A08 | Sphingosine Kinase 1 | 4.3E-5±1E-5 | 2.7E-5 *Pitman et al., 2015* | 1.6 |
| HA130 | Autotaxin | 4.2E-5±4.8E-5 | 2.8E-8 *Albers et al., 2010* | 1,522 |
| Ibuprofen | Cox-2 | 2.7E-4±6E-5 | 3.7E-4 *Noreen et al., 1998* | 0.73 |
| NS398 | Cox-2 | 1E-4±5E-5 | 3.8E-6 *Futaki et al., 1994* | 26.3 |
| NF-449 | GαS | 5.2E-5±3E-5 | 7.9E-6 *Hohenegger et al., 1998* | 6.5 |

because pristimerin showed a trend toward inhibition at the highest concentration, we tested its effect at low P4 concentration where it inhibited oocyte maturation but required a concentration of $10^{-5}$ M (*Figure 3—figure supplement 1C*), which is two orders of magnitude higher than its documented IC$_{50}$ in other systems (*Mannowetz et al., 2017*). These results argue against an important role for a monoacylglycerol enzymatic activity, whether ABHD2-dependent or not, in inducing oocyte maturation. We further tested a reported specific ABHD2 inhibitor, compound 183 ($10^{-4}$ M), which inhibited oocyte maturation at low and high P4 concentrations (*Figure 3—figure supplement 1D*), arguing for a role for ABHD2 in oocyte maturation.

We next focused on enzymes that generate PGs and LPA as the enriched lipid metabolites in response to P4 (*Figure 2D*). LPA can be produced through the action of PLA2 on phosphatidic acid (PA) or through autotaxin metabolism of LPE/LPC (*Figure 2D*). Blocking autotaxin activity with HA130 repressed oocyte maturation with an IC$_{50}$ of $4.2×10^{-5}$ M in the presence of high P4 (*Figure 3E*). We, therefore, tested the effects of HA130 at low P4, where it significantly inhibited maturation by ~80% (*Figure 3F*). However, this inhibition of maturation required high concentrations of HA130 -at least three orders of magnitude higher than the reported HA130 IC$_{50}$, arguing against a major role for the autotaxin pathway in generating LPA in response to P4, and rather supporting LPA production primarily through PLA2 activity.

We then tested the effect of inhibiting cyclooxygenases responsible for PGs production on oocyte maturation. Inhibition of Cox enzymes with ibuprofen effectively blocked maturation with an IC$_{50}$ of $2.7×10^{-4}$ M consistent with its IC$_{50}$ against Cox2 ($3.7×10^{-4}$ M) (*Figure 3E*, *Table 1*). As expected from its effect on maturation, ibuprofen treatment blocked both Plk1 and MAPK activation (*Figure 3G*). Specific inhibition of Cox2 using NS398 blocked maturation with an IC$_{50}$ of $10^{-4}$ M (*Figure 3E*), which is higher than its documented IC$_{50}$ against Cox2 ($3.8×10^{-6}$ M) (*Table 1*). Consistently, NS398 at high concentrations had a more pronounced inhibitory effect on oocyte maturation (~85%) at limiting P4 concentrations (*Figure 3H*). The Cox1-specific inhibitor SC-560 had no effect on oocyte maturation (*Figure 3E*). Collectively these data argue for a role for Cox2 in supporting maturation but not for Cox1.

Interestingly, inhibition of lipoxygenases using MK591 potentiated maturation by ~20% in the presence of high P4 (*Figure 3—figure supplement 1E*), and by 100% at low P4 (*Figure 3—figure supplement 1F*). Leukotrienes are decreased in response to P4 (*Figure 2*), so presumably inhibition of lipoxygenases diverts AA metabolism toward cyclooxygenase hydrolysis thus increasing PGs production, which would explain the observed effect on oocyte maturation.

## S1P signaling supports oocyte maturation

The lipidomics data show the downregulation of most metabolites along the sphingolipid arm of the metabolic pathway with a presumed enrichment of S1P (*Figure 2D*). We were, however, unable to detect an increase of S1P in the oocyte in response to P4 even when using a targeted mass spectrometry-based assay for S1P (*Figure 2—figure supplement 1D*). This may be because S1P is readily secreted and signals extracellularly. However, activation of ceramidases could be involved in the sphingolipid changes. mPRs share sequence and structural homology with adiponectin receptors, which have been shown to possess ceramidase activity that requires Zn in the active site that acts on the ceramide amide bond (*Holland et al., 2011*). We have previously shown that an mPRβ

mutant with all four zinc coordinating residues mutated to Ala to abrogate Zn binding (H129, D146, H281, H285A) is functional in inducing oocyte maturation (*Nader et al., 2020*). This argues that mPRβ does not require ceramidase activity to release oocyte meiotic arrest. To confirm this finding, we mutated Ser125 in the conserved SxxxH motif in mPRβ, which matches the ceramidase domain in PAQR and AdipoQ receptors (*Kelder et al., 2022*; *Tanabe et al., 2015*). The mPRβ-S125A mutant expresses and traffics normally to the cell membrane to similar levels as WT mPRβ (*Figure 4—figure supplement 1A*). Furthermore, it signals effectively downstream of progesterone as its overexpression (*Figure 4—figure supplement 1B*) rescues mPRβ knockdown with similar efficiency as WT mPRβ (*Figure 4—figure supplement 1C*), and activates Plk1, MAPK, and MPF to induce oocyte maturation (*Figure 4—figure supplement 1D*). Collectively these results argue against a role for mPRβ-dependent ceramidase activity in inducing oocyte maturation. Hence, changes in sphingolipids might be a consequence of the alterations in other lipid metabolic pathways, especially that the metabolism of GPLs is linked to that of sphingolipids through the activity of SM synthase.

If indeed S1P is the end product of sphingolipid metabolism in response to P4, then one would predict the activation of sphingosine kinases and concurrently ceramidases to support the metabolic flux. We tested this hypothesis using broad-spectrum sphingosine kinase and ceramidase inhibitors. Inhibition of ceramidase activity using D-e-MAPP had little effect on oocyte maturation at high P4 ($10^{-5}$ M) (*Figure 3I*). In contrast, at limiting P4 ($10^{-7}$ M) MAPP inhibited oocyte maturation by ~55% compared to untreated oocytes (*Figure 3J*). Note that at low P4 ($10^{-7}$ M) oocyte maturation is not complete as only the most primed oocytes respond, and this response varies from frog to frog (10–50% of the maturation observed with high P4). Inhibition of sphingosine kinases using MP-A08 inhibited oocyte maturation with an $IC_{50}$ of $4.3 \times 10^{-5}$ M at high P4 (*Figure 3I* and *Table 1*). At low P4, MP-A08 blocked maturation by 87.5% (*Figure 3K*), supporting a role for S1P in oocyte maturation. MP-A08 treatment primarily blocks Plk1 activation and to a lesser extent MAPK (*Figure 3L*).

To assess the role of S1P signaling in releasing oocyte meiotic arrest, we targeted S1P receptor (S1PR) isoforms based on their expression in the ovary. Of the five S1PR isoforms, only S1PR2 and S1PR3 are expressed in the ovary with higher expression of S1PR3 (*Massé et al., 2010*). We first confirmed the expression profile in oocytes using RT-PCR (*Figure 4A*). S1PR3 expression levels were higher than S1PR2 (*Figure 4A*). We confirmed the efficiency of the S1PR2 primers in the spleen where S1PR2 was shown to be highly expressed (*Massé et al., 2010*). We then focused on S1PR3 as the primary S1PR in the oocyte and knocked down its expression using antisense oligos. We validated S1PR3 knockdown at the RNA (*Figure 4B*) and protein (*Figure 4C*) levels. S1PR3 knockdown blocked oocyte maturation induced by either P4 or the mPR-specific agonist OD-02 by ~50% (*Figure 4D*).

Interestingly, S1PR3 knockdown did not affect P4-mediated activation of MAPK and Plk1 (*Figure 4E*), yet P4 was unable to dephosphorylate and activate MPF (*Figure 4E*). Plk1 induces maturation by activating the dual-specificity phosphatase Cdc25C, which dephosphorylates Cdc2 to activate MPF and is considered a rate-limiting step in MPF activation (*Perdiguero and Nebreda, 2004*). We, therefore, tested the effects of S1PR3 knockdown on Cdc25C activation and showed that despite the induction of both Plk1 and MAPK, Cdc25C was not activated in response to P4 in S1PR3 knockdown cells (*Figure 4F*). We followed Cdc25C activation by assessing its dephosphorylation at Ser287 (Ser216 in humans), which is required for its activation (*Figure 4F*; *Perdiguero and Nebreda, 2004*). The partial oocyte maturation block in S1PR3 KD oocytes argues for a modulatory role for S1P signaling. We tested this hypothesis by assessing the requirement for S1PR3 at increasing P4 dosages that would incrementally induce the multiple arms of the maturation signaling pathways. S1P3R knockdown oocytes matured with significantly less efficiency and with a slower time course at low P4 ($10^{-7}$ M) (*Figure 4G*), but interestingly this effect was gradually lost with increased P4 concentrations ($10^{-5}$ and $10^{-4}$ M) (*Figure 4G*). These data argue that the S1P/S1PR3 pathway modulates oocyte maturation, but that by itself it is not essential for maturation as it can be bypassed when other pathways are hyper activated at high P4 concentrations. This is consistent with the production of multiple lipid messengers to induce maturation through several GPCR pathways.

## Role of G-protein coupled receptors (GPCRs) in mediating lipid messenger action

All the enriched lipid metabolites detected from our lipidomics analyses - PGs, LPA, and possibly S1P - act through GPCRs. This raises the intriguing possibility that P4 expands its signaling through the

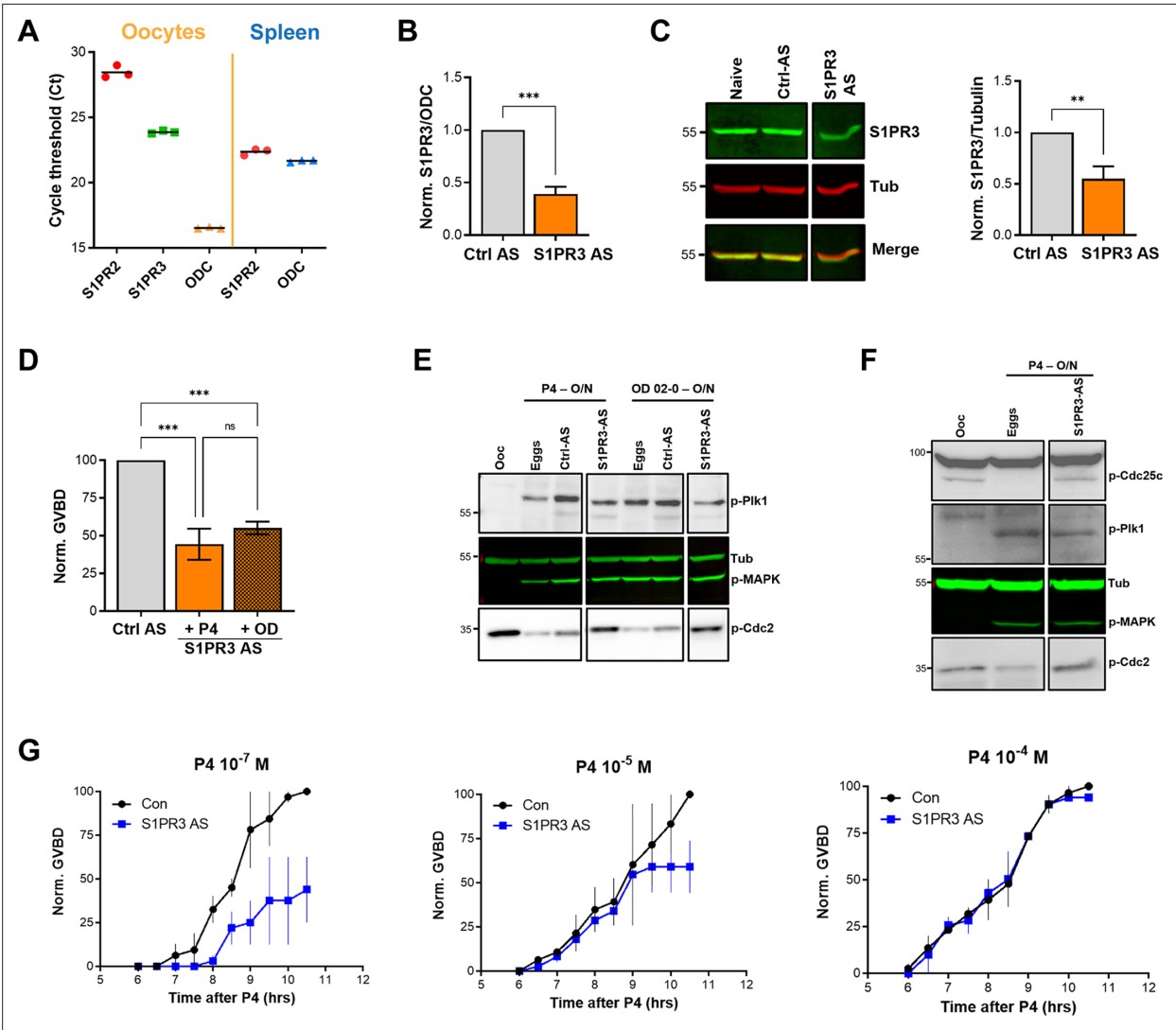

**Figure 4.** S1P signaling and oocyte maturation. (**A**) mRNAs levels of S1PR2, S1PR3, and the housekeeping gene Ornithine decarboxylase (ODC) in oocytes and spleen measured as the Cycle threshold (Ct) from real-time PCR. (**B–C**) S1PR3 knockdown. Oocytes were injected with control antisense (Ctrl AS) or specific S1PR3 antisense oligonucleotides and incubated at 18 °C for 24 hr. RNAs and protein extracts were prepared and analyzed by RT-PCR and western blot (WB) to determine the efficacy of S1PR3 knockdown as compared to control oocytes (Ctrl AS). B. Histogram showing the relative RNAs levels of S1PR3 mRNA to xODC. (**C**) Representative WB (left panel) and normalized quantification (right panel) comparing S1PR3 protein levels between naïve, Ctrl AS and S1PR3 AS injected oocytes. Tubulin is shown as a loading control (mean ± SEM; n=6 independent female frogs). (**D**) Oocyte maturation following injection of S1PR3 antisense, normalized to progesterone (P4) or Org OD 02–0 (OD)-treated oocytes injected with control antisense (Ctrl AS) (mean ± SEM; n=7 independent female frogs, ordinary one-way ANOVA). (E/F) Representative WBs of MAPK, Plk1, and Cdc2 (**E**), as well as CDC25C (**F**) phosphorylation from untreated oocytes, P4 or OD, matured eggs (O/N)D, oocytes injected with control antisense (Ctrl AS) or S1PR3 antisense (S1PR3 AS) and treated O/N with P4 or OD. Tubulin is shown as a loading control.(**G**) GVBD-time course after treatment with P4 at the indicated concentrations in oocytes injected with water (Con) or with S1PR3 antisense (S1PR3 AS) (mean ± SEM; n=2 independent female frogs).

The online version of this article includes the following source data and figure supplement(s) for figure 4:

**Source data 1.** Original files for western blot analysis are displayed in *Figure 4C,E,F*.

**Source data 2.** File containing labeled western blots for *Figure 4C,E,F*, indicating the relevant bands and treatments.

**Figure supplement 1.** Role of GPCR signaling.

**Figure supplement 1—source data 1.** Original files for western blot analysis are displayed in *Figure 4—figure supplement 1B,D,F*.

**Figure supplement 1—source data 2.** File containing labeled western blots for *Figure 4—figure supplement 1B,D,F*, indicating the relevant bands and treatments.

production of lipid messengers that act on several GPCRs to mediate the multiple cellular changes that need to occur concurrently during oocyte maturation. This is an attractive possibility as GPCRs have been implicated in *Xenopus* oocyte maturation for decades without conclusive evidence for their involvement, and they have been shown to mediate mPR nongenomic signaling. Therefore, we tested the effects of broad inhibitors of trimeric G-proteins. Blocking $G\alpha_s$ using NF-449 partially inhibited maturation ($IC_{50}$ $5\times10^{-5}$ M) (*Figure 4—figure supplement 1E*), whereas blocking either $G\beta\gamma$ or $G\alpha_{q/11}$ with gallein and YM-254890, respectively was inefficient at blocking oocyte maturation (*Figure 4—figure supplement 1E*). Interestingly, inhibiting $G\alpha_s$ with NF449 primarily blocked Plk1 activation to inhibit MPF and oocyte maturation, and to a lesser extent the MAPK pathway (*Figure 4—figure supplement 1F*). At low P4 concentration, the inhibitory effect of NF-449 was still observed as well as inhibition by gallein but not YM-254890 (*Figure 4—figure supplement 1G*). This argues for a supporting role for $G\beta\gamma$ signaling as well in oocyte maturation. These results support a role for $G\alpha_s$ signaling and potentially $G\beta\gamma$ downstream of P4 through lipid intermediates that require PLA2, Cox2, and SphK activities.

## ABHD2 is a PLA2 that requires mPRβ as a co-receptor

Both the metabolomics and pharmacological data strongly implicate PLA2 activation downstream of P4 to induce maturation. In addition, several lines of evidence argue that this PLA2 activity requires ABHD2: (1) Induction of the signaling cascades downstream of P4 requires both ABHD2 and PLA2; (2) the lipidomics changes downstream of P4 are dependent on both activities; (3) mutations in the ABHD2 lipase domain abolish its ability to induce maturation; (4) ABHD2 has been characterized as a lipase acting on either MAG or TAG (*Miller et al., 2016*; *Naresh Kumar et al., 2016*); and (5) an ABHD2 specific inhibitor (compound 183) inhibits maturation. Collectively, these findings suggest that ABHD2 is associated with a PLA2 activity that is induced in response to P4 to release oocyte meiotic arrest.

To test this hypothesis, we translated ABHD2 and mPRβ in vitro in rabbit reticulocyte lysates and assessed whether they possess PLA2 activity. Expression of ABHD2 or mPRβ alone was not associated with any increase in PLA2 activity whether in the presence or absence of P4 (*Figure 5A*). Interestingly though, co-expression of both mPRβ and ABHD2 produced a PLA2 activity, but only in the presence of P4 (*Figure 5A*). This is despite our inability to detect either mPR or ABHD2 expression by Western blots from the reticulocyte lysates reactions. These results argue that both mPRβ and ABHD2 are required for the P4-dependent PLA2 activity, but they do not address which receptor is the catalytic subunit. However, our functional experiment with the ABHD2 hydrolase domain mutants (S/D/H and S207) strongly argues that ABHD2 is the PLA2 catalytic subunit.

To directly confirm this prediction, we co-expressed mPRβ with either ABHD2 WT or the S/D/H mutant in reticulocyte lysates and tested their PLA2 activity. Mutating the ABHD2 lipid hydrolase domain (S/D/H mutant) resulted in a loss of PLA2 activity when the protein was expressed with mPRβ in the presence of P4 (*Figure 5B*). Taken together, these results argue that ABHD2 has PLA2 catalytic activity but only in the presence of mPRβ.

To cross-validate our findings from the reticulocyte experiments, we tested in vitro and cell-based expression systems to increase protein yield. We settled on the coupled in vitro transcription/translation tobacco lysates system (ALiCE) that produces high protein yields and targets expressed proteins to microsomes directly using a melittin signal peptide. This is an advantage for our proteins of interest as they are both integral membrane proteins. The tobacco lysates expression system produced significantly higher yields of both mPR and ABHD2 expression where both proteins were readily detected on Western blots (*Figure 5C*). Similar to the reticulocyte system, tobacco lysates expressing ABHD2 or mPRβ alone were not associated with increased PLA2 activity whether in the presence or absence of P4 (*Figure 5D*). Of note tobacco lysates have endogenous PLA2 activity, which was inhibited by expression of mPRβ alone (*Figure 5D*). The PLA2 activity data from tobacco lysates are reported as the rate of increase over time (*Figure 5D*). Co-expression of mPRβ and ABHD2 in the tobacco lysates produced significant PLA2 activity that was not only higher than that detected from the reticulocyte lysates but also stable over extended periods of time allowing recordings for over 2 hr (*Figure 5D*). Interestingly in this case, however, the PLA2 activity of the co-receptor complex was independent of P4 (*Figure 5D*). Both the reticulocyte and tobacco expression system show that mPRβ is a necessary partner for ABHD2 to mediate its PLA2 activities. However, the reticulocyte co-expression results

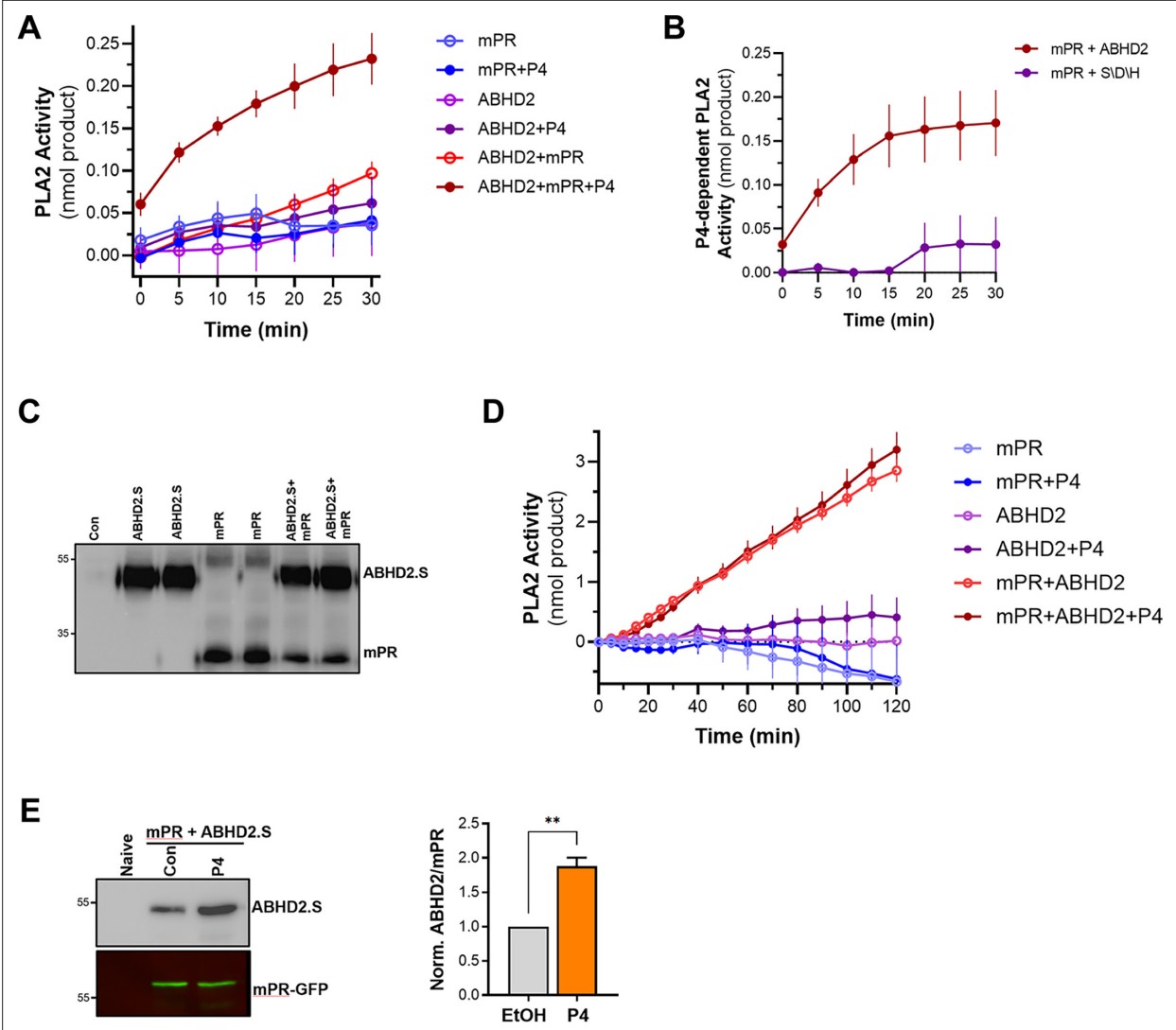

**Figure 5.** Phospholipase A2 (PLA2) activity of the membrane progesterone receptor β (mPRβ)-α/β hydrolase domain-containing protein 2 (ABHD2) co-receptor complex. (**A**) Time-dependent PLA2 activity from reticulocyte lysates expressing mPR, ABHD2.S, or ABHD2.S+mPR in the presence of ethanol as vehicle or P4 $10^{-5}$ M (mean ± SEM; n=3). (**B**) Time course of progesterone (P4)-dependent PLA2 activity in reticulocyte lysates expressing mPRβ with either wild-type ABHD2 (ABHD2) or the ABHD2 S207A/D345A/H376A mutant (S/D/H). P4-dependent PLA2 is plotted as the difference in activity in the presence and absence of P4 (Mean ± SEM; n=3). (**C**) Example western blot (WB) probed with anti-His antibody from tobacco lysates (ALiCE) alone (Con) and lysates expressing mPRβ or ABHD2 alone or both proteins in duplicated as indicated. (**D**) Time-dependent PLA2 activity from ALiCE lysates overexpressing mPR, ABHD2.S, or ABHD2.S+mPR in the presence of the vehicle ethanol or P4 $10^{-5}$ M (mean ± SEM; n=3). Data are plotted as the rate of production of the lysothiophospholipid product from the beginning of the experiment (0 min) with the rate of the lysates alone subtracted. (**E**) Representative immunoprecipitation (IP) WB and quantification of mPR-GFP from oocyte lysates from un-injected oocytes (Naive) or oocytes over-expressing mPR-GFP and ABHD2.S treated for 40 min with Ethanol (EtOH) or P4. *Left panel,* the representative WB membrane is probed for ABHD2 and GFP (mean ± SEM; n=3 independent female frogs, unpaired t-test).

The online version of this article includes the following source data and figure supplement(s) for figure 5:

**Source data 1.** Original files for western blot analysis are displayed in *Figure 5C,E*.

**Source data 2.** File containing labeled western blots for *Figure 5C,E*, indicating the relevant bands and treatments.

**Figure supplement 1.** ABHD2 mutants interaction with mPRβ.

**Figure supplement 1—source data 1.** Original files for western blot analysis are displayed in *Figure 5A*.

**Figure supplement 1—source data 2.** File containing labeled western blots for *Figure 5A*, indicating the relevant bands and treatments.

argue that mPRβ and ABHD2 need P4 to assemble and form a complex to be able to mediate the ABHD2 PLA2 activity. In contrast, when the two proteins are expressed at high levels and more importantly targeted specifically to the same microsomal compartment their PLA2 activity becomes P4 independent. We were thus interested in testing whether the two receptors interact in vivo. We expressed mPRβ-GFP in oocytes and tested whether it forms a complex with ABHD2. Immunoprecipitation of mPRβ-GFP pulls down ABHD2 at rest (*Figure 5E*), and importantly this interaction is significantly enhanced (1.9±0.2 fold, p=0.002) following P4 treatment (*Figure 5E*). This argues that P4 enhances the assembly of the co-receptor complex. Furthermore, co-expression and pull-down experiments show that the non-functional ABHD2 mutants: S/D/H and ABHD2.L interact with mPRβ (*Figure 5—figure supplement 1A*), indicating that the interaction between ABHD2 and mPRβ does not require a functional lipase domain.

## ABHD2 and PLA2 activity are required for mPRβ endocytosis and signaling

We have previously shown that mPRβ induces oocyte maturation at the level of the signaling endosome following its clathrin-dependent endocytosis (*Nader et al., 2020*). Importantly, mPRβ endocytosis is sufficient to induce maturation in the absence of P4. Since changes in plasma membrane lipid composition are known to modulate endocytosis and membrane component activities, we suspected a role for ABHD2/PLA2 in mPRβ endocytosis in response to P4. Therefore, we quantified mPRβ endocytosis following ABHD2 knockdown. P4 leads to the enrichment of mPRβ positive intracellular vesicles, and this enrichment requires ABHD2 as it was lost in ABHD2 knockdown oocytes (*Figure 6A*). PLA2 activity was also critical for P4-induced mPRβ endocytosis, which was completely blocked in ACA-treated oocytes (*Figure 6B*). Interestingly, ACA also inhibited basal endocytosis of mPRβ in the absence of P4, which is apparent as lower vesicular mPRβ in oocytes treated with ACA (*Figure 6B*, no P4). This argues that both ABHD2 and PLA2 are required for mPRβ endocytosis and by extension its signaling in response to P4.

We were, therefore, interested in assessing the subcellular distribution of ABHD2 during oocyte maturation to determine whether it is also internalized. We thus tagged it at the N- or C-terminus with mCherry. Unfortunately, both mCh-tagged ABHD2s were not functional as they did not rescue ABHD2 knockdown (*Figure 5—figure supplement 1B*), despite efficient expression and binding to mPR (*Figure 5—figure supplement 1A*). This prevented us from imaging the trafficking and distribution of ABHD2 during oocyte maturation.

The requirement for ABHD2 to mediate P4-dependent mPRβ endocytosis (*Figure 6A*) suggests that this may be the primary function of ABHD2 in response to P4. Under that scenario, P4 activates ABHD2 enzymatic activity which induces mPRβ endocytosis leading to oocyte maturation. To test this hypothesis, we were interested in inducing mPRβ endocytosis in the absence of P4. We have previously shown that a dominant negative SNAP25 mutant missing the last 20 residues (SNAP25Δ20), effectively blocks exocytosis in the oocyte (*Figure 6C*); leading to mPRβ enrichment intracellularly and inducing maturation in the absence of P4 (*Nader et al., 2020*; *El-Jouni et al., 2007*). Similar to P4, SNAP25Δ20-induced oocyte maturation requires clathrin-dependent endocytosis as it was blocked by a clathrin blocker pitstop (*Figure 6D*). As expected, SNAP25Δ20-induced oocyte maturation requires mPRβ as it was inhibited in oocytes where mPRβ was knocked down (*Figure 6E*). Surprisingly, SNAP25Δ20-induced maturation also requires ABHD2, since oocytes where ABHD2 is knocked down were unable to mature in response to SNAP25Δ20 (*Figure 6E*), despite high expression of SNAP25Δ20 (*Figure 6F*). We further tested whether mPRβ overexpression allows ABHD2 KD oocytes to mature in response to SNAP25Δ20, but this was not the case (*Figure 6E*). This argues that even in the presence of excess mPRβ, its forced internalization using SNAP25Δ20 (*Figure 6C*) in the absence of ABHD2 is not sufficient to induce maturation. Therefore, ABHD2 activity is not only required for mPRβ internalization but it is also required at the level of the signaling endosome following mPRβ endocytosis to induce maturation.

Like P4, SNAP25Δ20 activates Plk1, MAPK, and MPF (*Figure 6G*) and this induction was significantly inhibited following the knockdown of either ABHD2 or mPRβ (*Figure 6G and H*), showing that both proteins signal through the canonical kinase cascades to induce maturation following internalization even in the absence of P4. Furthermore, in addition to the PLA2 inhibitor ACA, inhibition of Cox2, or sphingosine kinase partially blocked SNAP25Δ20 induced maturation with ACA being the most

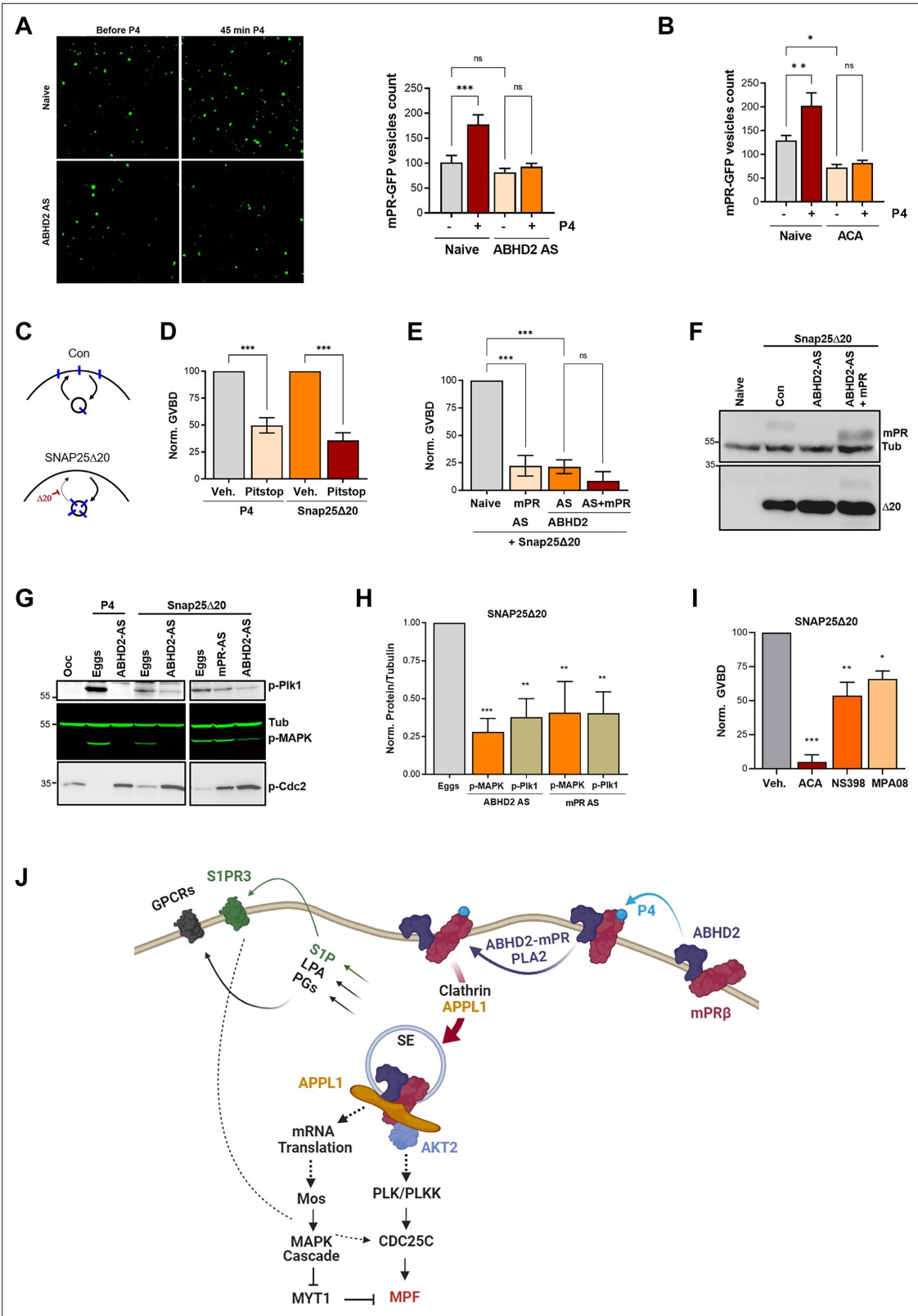

**Figure 6.** α/β hydrolase domain-containing protein 2 (ABHD2) and phospholipase A2 (PLA2) activity are required for membrane progesterone receptor β (mPRβ) endocytosis and signaling. (**A**) Effect of ABHD2 knockdown on mPR-P4 mediated endocytosis. Oocytes were injected with mPR-GFP RNA (Naïve) and 48 hr later either left untreated or injected with ABHD2 antisense (ABHD2 AS). The following day oocytes were imaged, before and 45 min after progesterone (P4) treatment. *Left panel,* representative confocal image of mPR-GFP positive vesicles in naïve and ABHD2 AS oocytes, before

*Figure 6 continued on next page*

*Figure 6 continued*

and after P4. *Right panel,* histogram of mPR-GFP positive vesicle count before and after P4 (mean ± SEM; n=20–21 oocytes per condition, from three independent female frogs, ordinary one-way ANOVA). (**B**) PLA2 inhibition blocks P4-mediated endocytosis. Vesicle count from oocytes expressing mPR-GFP for 72 hr was pretreated with vehicle (naïve) or ACA for 2 hr, followed by imaging, before and 45 min after P4 treatment (mean ± SEM; n=16–18 oocytes per condition, from two independent female frogs, ordinary one-way ANOVA). (**C**) Cartoon depicting the role of SNAP25Δ20 in blocking exocytosis. (**D**) SNAP25Δ20-induced oocyte maturation requires clathrin-dependent endocytosis. Naïve oocytes were pretreated with vehicle or Pitstop ($10^{-5}$ M), followed by overnight treatment with P4 or SNAP25Δ20-mRNA injection. Oocyte maturation in P4 or SNAP25Δ20 injected oocytes normalized to the vehicle condition (mean ± SEM; n=4 independent female frogs, ordinary one-way ANOVA). (**E**) SNAP25Δ20-induced maturation requires ABHD2. Oocytes were injected with mPR antisense (mPR AS) or ABHD2 antisense (AS) with or without mPRβ mRNA (AS +mPR). 48 hr later, oocytes were injected with mRNA to overexpress SNAP25Δ20. The following day, oocyte maturation was measured in mPR AS, ABHD2 AS, and ABHD2 AS +mPR oocytes normalized to naive oocytes injected with SNAP25Δ20 (mean ± SEM; n=3 independent female frogs, ordinary one-way ANOVA). (**F**) Representative WB of mPR and SNAP25Δ20 proteins expression in naïve, ABHD2 AS and ABHD2 AS +mPR oocytes. Tubulin is shown as a loading control. (**G**) Representative WB of MAPK, Plk1, and Cdc2 phosphorylation from untreated oocytes, P4 matured eggs, or SNAP25Δ20 mRNA injection, and oocytes injected with mPR (mPR AS) or ABHD2 antisense (ABHD2 AS) and treated O/N with P4 or SNAP25Δ20 mRNA injection. Tubulin is shown as a loading control. (**H**) Quantification of p-Plk as the ratio of p-PLK/Tubulin and p-MAPK as the ratio of p-MAPK/Tubulin normalized to the ratios in naive eggs (mean ± SEM; n=4 independent female frogs, ordinary one-way ANOVA). (**I**) Oocyte maturation in oocytes pretreated for 2 hr with ACA, NS398, and MP-A08 followed by SNAP25Δ20-mRNA injection and normalized to GVBD in oocytes pre-treated with vehicle followed by SNAP25Δ20-mRNA injection (Veh.) (mean ± SEM; n=3 independent female frogs, ordinary one-way ANOVA). (**J**) Model of the signaling cascade triggered in response to P4 (generated using Biorender).

The online version of this article includes the following source data for figure 6:

**Source data 1.** Original files for western blot analysis are displayed in *Figure 6F,G*.

**Source data 2.** File containing labeled western blots for *Figure 6F,G*, indicating the relevant bands and treatments.

effective (*Figure 6I*), similar to what we observe with P4. ACA blocks basal mPRβ recycling (*Figure 6B*), and as such would inhibit its internalization in response to SNAP25Δ20 expression. However, the Cox2 (NS398) and the SphK (MPA08) inhibitors should not interfere with mPRβ internalization in response to SNAP25Δ20, although experimentally it is difficult to ascertain this. Collectively these results argue that PLA2 activity is required for mPRβ internalization and that furthermore, both Cox2 and SphK activities are important to modulate oocyte maturation following mPRβ internalization.

## Discussion

mPR-dependent nongenomic signaling is an important regulator of many physiological processes in female and male reproduction, cardiovascular, neuroendocrine, neurological, and immune function (*Dressing et al., 2011*; *Valadez-Cosmes et al., 2016*; *Moussatche and Lyons, 2012*; *Wendler and Wehling, 2022*). This raises interest in mPRs as potential therapeutic targets for hypertension and other cardiovascular diseases, reproductive disorders, neurological diseases, and cancer (*Wendler and Wehling, 2022*). Therefore, understanding mPR signaling is important to assess their physiological and pathological contributions. In this study, we used oocyte maturation in the frog as a well-established model for P4 nongenomic signaling.

Oocyte maturation prepares the egg for fertilization and endows it with the ability to activate and initiate embryonic development (*Machaca, 2007*). As such it represents the initial step in multicellular organismal development that precedes fertilization. It is, therefore, not surprising that oocyte maturation involves multiple signaling cascades to mediate both the completion of meiosis -the so-called nuclear maturation- to generate a haploid gamete, and the extensive cellular differentiation of the oocyte that is needed to support development, and includes protein synthesis, membrane remodeling, and $Ca^{2+}$ signaling differentiation that supports the block of polyspermy and resumption of meiosis at fertilization (*Machaca, 2007*).

In *Xenopus* oocytes, the release of the long-term meiotic arrest can be triggered by P4. In fact, *Xenopus* oocyte maturation is one of the most studied examples of nongenomic P4 signaling and represents a well characterized system to define the signaling cascade downstream of P4. It is well-established that P4 in the oocyte activates two parallel and interdependent kinase cascades that ultimately culminate in MPF activation (*Nebreda and Ferby, 2000*). Despite many studies over the past decades, the earliest signaling steps downstream of mPR have remained elusive. An interesting recent development shows that clathrin-dependent endocytosis of mPRβ is necessary and sufficient for its signaling and the induction of oocyte maturation even in the absence of P4 (*Nader et al., 2020*).

It, therefore, appears that the signaling endosome acts as a hub to activate the multiple pathways required to release meiotic arrest, resume meiosis, and prepare the egg for fertilization.

An untested assumption is that P4 induces oocyte maturation primarily through mPRβ since its knockdown or anti-mPRβ antibodies block maturation (*Nader et al., 2020*; *Josefsberg Ben-Yehoshua et al., 2007*). However, the oocyte expresses other P4 receptors including ABHD2 that may contribute to P4 signaling. Here, we show that knockdown of ABHD2 blocks oocyte maturation and that this is rescued effectively by ABHD2 overexpression but not by mPRβ overexpression. ABHD2 contains a lipase and an acyltransferase motif. The lipase domain is required for oocyte maturation but not the acyltransferase domain. Based on this finding, we undertook an unbiased lipidomics approach to better define metabolic alterations driven by ABHD2 lipase activity. P4 resulted in a broad decrease in glycerophospholipid and sphingolipid species, with the enrichment of a few lipid messenger end products, namely prostaglandins, LPA, and S1P (*Figure 6J*). This is intriguing as all these lipid messengers act through GPCRs. We in fact validate S1P action through the S1P receptor, which is a GPCR, to support oocyte maturation. These findings argue that P4 activates ABHD2 lipase activity to generate lipid messengers that in turn stimulate various GPCRs that regulate the multiple aspects of oocyte maturation. This would nicely explain the standing controversy in the field as to whether mPRs are GPCRs or not. Should the findings from the oocyte hold in other systems, it would be argued that P4 activates lipid catabolism to generate lipid messenger that then acts through GPCRs. Thus, nongenomic P4 signaling would involve both non-GPCR and GPCR signaling modalities in series.

An intriguing finding of the current study is that the PLA2 activity of ABHD2 is activated through its interaction with mPRβ (*Figure 6J*). Although ABHD2 and mPRβ interact at rest, this interaction is enhanced following P4 treatment. This is the first report showing that ABHD2 possesses PLA2 activity and that this PLA2 activity requires interaction with mPRβ. The PLA2 family is large and complex with over 50 members that have been classified based on enzymatic activity, co-factor dependence, subcellular distribution, and structure (*Dennis et al., 2011*; *Murakami et al., 2020*); and include the secreted, cytosolic, $Ca^{2+}$-independent, lysosomal, platelet-activating factor acyl hydrolase, and some members of the α/β hydrolase domain (ABHD) family (*Lord et al., 2013*).

ABHD2 has been shown to act as a 2-AG lipase to activate sperm (*Miller et al., 2016*; *Mannowetz et al., 2017*). Interestingly, we find no evidence for such an activity in response to P4 based on our lipidomics analyses in the oocyte. This raises the interesting possibility that ABHD2 lipase specificity may be regulated by protein-protein interactions in a cell-specific fashion, but this remains to be tested.

In addition to the generation of lipid messengers that branch out P4 signaling at its onset into multiple signaling cascades through GPCRs, the ABHD2 PLA2 activity hydrolyzes phospholipids to generate AA that is then metabolized to PGs through the activity of Cox2. The hydrolysis of membrane phospholipids generates lysophospholipids with a single acyl chain that tends to have a conical shape. Such conical phospholipids are prone to inducing membrane curvature (*Harayama and Riezman, 2018*) as the latter is determined by the size of the lipid headgroups and hydrophobic tails (*Ernst et al., 2016*). Conical lipids have been shown to play important roles in membrane fusion events both in vitro and in vivo (*Zick et al., 2014*; *Pagliuso et al., 2016*; *Irie et al., 2017*). It is, therefore, tempting to postulate that enrichment in lysophospholipids following ABHD2-PLA2 activation generates spontaneous membrane curvature, which can be sensed by the clathrin endocytic machinery to induce mPRβ endocytosis and initiate signaling and maturation (*Figure 6J*).

Collectively our findings define the earliest steps in nongenomic P4 signaling to release the oocyte meiotic arrest and prepare the egg for fertilization (*Figure 6J*). Furthermore, they show for the first time that two heterologous receptors previously implicated in nongenomic signaling, mPRβ and ABHD2, function as coreceptors to induce PLA2 activity that is required for both endocytosis and further downstream signaling. Nongenomic P4 signaling is widespread and mediates reproductive, neurological, and other functions. Therefore, these findings are likely to have broad implications throughout biology.

## Materials and methods
### Reagents and primers
A list of the antibodies, siRNA, chemicals, and other reagents used is provided in *Supplementary file 1a*. Primers and sense/antisense oligos used are listed in *Supplementary file 1b*.

## *Xenopus laevis* oocytes

All animal procedures and protocols were performed in accordance with the University of Weill Cornell Medicine-Qatar Institutional Animal Care and Use Committee. *Xenopus laevis* female frogs were obtained from *Xenopus* I (adult females Lab bred). Stage VI oocytes were obtained as previously described (*Machaca and Haun, 2002*). Oocytes were maintained in L-15 medium solution (Sigma-Aldrich Cat# L4386) supplemented with HEPES (Sigma-Aldrich Cat# H4034), 0.1% (v/v) of penicillin/streptomycin stock solution (Thermo Fisher Scientific Cat# 15140–122) and 0.1% (v/v) of gentamycin (EMD Millipore Cat# 345814–1 GM) at pH 7.6. The oocytes were used 24–72 hr after harvesting and injected with RNAs or sense/antisense oligos and kept at 18 °C for 1–2 d to allow for protein expression or knockdown. After treatment with progesterone at $10^{-7}$ M, $10^{-5}$ M, or $10^{-4}$ M (as indicated), GVBD was detected visually by the appearance of a white spot at the animal pole. For the inhibitor studies, oocytes were preincubated for 2 hr with different concentrations of the inhibitors, followed by overnight incubation with P4 at $10^{-7}$ M or $10^{-5}$ M concentration. Co-immunoprecipitation, Western blot, and oocyte imaging are described in supplemental data.

## Molecular biology

Generation of pSGEM-mPRβ-GFP, pSGEM-TMEM-mCherry, and pSGEM-SNAP25Δ20 were previously described (*Nader et al., 2020* and El jouni et al.2007). To introduce the serine to alanine mutation at amino acid 125 (S125A) within the zinc-binding domains in pSGEM-mPR-GFP, the XL QuikChange mutagenesis kit (Agilent Technologies) was used. Coding sequences for ABHD2.L (XB-GENE-6489069) and ABHD2.S (XB-GENE-17345096) encoding the *Xenopus* ABHD2.L and S, respectively were synthetized tagged or not with mCherry and cloned in pSGEM by Gene universal. Mutants mPR S125A, ABHD2 H120A, and N125A were generated using the XL QuikChange mutagenesis kit. The rest of the ABHD2 mutants were generated by Gene universal. All constructs were verified by DNA sequencing and by analytical endonuclease restriction enzyme digestion. mRNAs for all the pSGEM clones were produced by in vitro transcription after linearizing the vectors with *Nhe*I, using the mMessage mMachine (Ambion). Relative expression of mPR, ABHD2 LS, ABHD2.S, S1PR2, and S1PR3 were assessed by quantitative real-time PCR (Affymetrix), with *Xenopus* Ornithine decarboxylase (xODC) as the internal control to normalize mRNA transcript levels (*Sindelka et al., 2006*), after total RNA extraction using the RNeasy mini Kit (Qiagen).

## Co-immunoprecipitation

Around 70 Oocytes overexpressing GFP-tagged proteins were lysed in IP solution (30 mM HEPES, 100 mM NaCl, pH 7.5) containing protease and phosphatase inhibitors (5 µl/ oocyte). Lysates were cleared of yolk by centrifugation at 1000×g three times for 10 min each at 4 °C. Supernatants were then solubilized with 4% NP40 for 1 hr followed by 15 min centrifugation at 18,188 ×g at 4 °C before immunoprecipitation using anti-GFP microbeads (1 µl/oocyte) per the manufacturer's instructions.

## Western blotting

Generally, cells were lysed in MPF lysis buffer 0.08 M β-glycerophosphate, 20 mM Hepes (pH 7.5), 15 mM MgCl$_2$, 20 mM EGTA, 1 mM Na-Vanadate, 50 mM NaF, 1 mM DTT, in the presence of protease and phosphatase inhibitors, followed by centrifugation 3 x at 1000 ×g for 10 min at 4 °C to remove yolk granules. For the treatment with the Calf intestinal phosphatase (CIP), oocytes were lysed in IP solution (30 mM HEPES, 100 mM NaCl, pH 7.5) (5 µl/ oocyte) containing protease and phosphatase inhibitors, and centrifuged twice at 1000×g for 10 min each at 4 °C. Lysates were separated on 10% SDS-PAGE gels, transferred to polyvinylidene difluoride (PVDF) membranes (Millipore), blocked for 1 hr at room temperature with 5% Milk in TBS-T buffer (150 mM NaCl and 20 mM Tris; pH 7.6, 0.1% Tween) and then incubated overnight at 4 °C in 3% BSA in TBS-T with the primary antibody. Blots were then washed three times with TBS-T and probed for 1 hr with horseradish peroxidase (HRP)-conjugated secondary antibody 1/10,000 (for ABHD2, SNAP25, GFP, His, p-plk1, p-cdc25, and p-cdc2), or with infrared fluorescence, IRDye 800 and 680 secondary antibodies (1/10,000) (for GFP, S1PR3, p-MAPK and Tubulin). The blots were visualized using ECL-based detection of horseradish peroxidase (HRP) followed by Image J analysis or using the LI-COR Odyssey Clx Infrared Imaging system and analyzed using LI-COR image Studio Lite v.4.0. The primary antibodies used are anti-GFP (1/1000), anti-SNAP25 (1/1000), anti-Tubulin (1/10,000), anti-p-plk1 (1/1000), anti-phospho-MAPK

(1/4000), anti-phospho-Cdc2 (1/1000), anti-phospho Cdc25 (1/1000), anti-His (1/1000), and anti-S1PR3 (1/1000).

## Oocytes imaging

Oocytes were imaged on a LSM880 confocal (Zeiss, Germany) fitted with a Plan Apo 63 x/1.4 oil immersion objective, with Z-stacks taking in 0.5 µm sections using a 1 Airy unit pinhole aperture. Images were analyzed using the ImageJ software. To measure the distribution of mPR at the cell membrane, TMEM was used as membrane marker. For each oocyte, the percentage (%) of membrane mPR was calculated by analyzing the intensity of fluorescence distribution through a z-stack of images, where we conservatively used two z stacks below the peak of WGA fluorescence, as a reference to mark the end of the plasma membrane compartment. For mPR-positive vesicle count, z-stacks of images from oocytes overexpressing mPR-GFP were collected before and after P4 in the presence or absence of ABHD2 antisense, or after 2 hr pretreatment with ACA. ImageJ software was used to quantify mPR vesicles using the 3D objects counter application.

## Metabolomic analyses

Untargeted metabolomics was performed on single oocytes or a group of 10 oocytes per condition. For measurements of metabolites in a group of 10 oocytes, three experiments using three independent female frogs were performed. Each experiment was conducted in 5 sample replicates per condition per frog. Metabolites were analyzed on both the Metabolon HD4 and CLP platforms. Tandem LC MSMS was used to measure prostaglandins and S1P from 10 oocytes using 10 replicates per condition. Levels of Eicosanoids, EETs, and HETEs before and 30 min after P4 treatment were measured in 20 single oocytes per condition by Cayman chemicals A detailed description of the techniques used is reported below.

## Sample preparation for metabolomics analysis

The untargeted metabolomic study was done on a single oocyte or a group of 10 oocytes per condition. For the single oocyte approach, individual oocyte was kept in 96 wells plate in 50 µl Ringer Buffer (96 mM NaCl, 2.5 mM KCl, 1.8 mM CaCl2, 2 mM MgCl2, 10 Mm Hepes, pH 7.4) (with or without $10^{-5}$ M Progesterone treatment). Oocytes from each condition (10 per condition) were collected individually in a 1.5 ml tube. Each oocyte was washed in 50 µl metabolite grade water three times, following aspiration after each wash to remove as much water as possible, without sucking up the oocyte. Washed oocytes were then flash-frozen on dry ice. The samples were kept at –80 °C until shipment for metabolite measurements.

Single oocyte was extracted in 80% MeOH (LC-MS grade methanol, Fisher Scientific) by bead-beating for 45 s using a Tissuelyser cell disrupter (Qiagen). Extracts were centrifuged for 5 min at 5000 rpm to pellet insoluble protein and supernatants were transferred to clean tubes. The extraction procedure was repeated two additional times, and all three supernatants were pooled, dried in a Vacufuge (Eppendorf), and stored at –80 °C until analysis. The methanol-insoluble protein pellet was solubilized in 0.2 M NaOH at 95 °C for 20 min and protein was quantified using a BioRad DC assay. On the day of metabolite analysis, dried cell extracts were reconstituted in 70% acetonitrile at a relative protein concentration of 4 µg/ml, and 4 µl of this reconstituted extract was injected for LC/MS-based untargeted metabolite profiling. LC/MS metabolomics platform for untargeted metabolite profiling.

Metabolite extract from a single oocyte was analyzed by LC/MS as described previously (*Chen et al., 2018*), using a platform comprised of an Agilent Model 1290 Infinity II liquid chromatography system coupled to an Agilent 6550 iFunnel time-of-flight MS analyzer. Chromatography of metabolites utilized aqueous normal phase (ANP) chromatography on a Diamond Hydride column (Microsolv). Mobile phases consisted of: (A) 50% isopropanol, containing 0.025% acetic acid, and (B) 90% acetonitrile containing 5 mM ammonium acetate. To eliminate the interference of metal ions on chromatographic peak integrity and electrospray ionization, EDTA was added to the mobile phase at a final concentration of 5 µM. The following gradient was applied: 0–1.0 min, 99% B; 1.0–15.0 min, to 20% B; 15.0–29.0, 0% B; 29.1–37 min, 99% B. Raw data were analyzed using MassHunter Profinder 8.0 and MassProfiler Professional (MPP) 15.1 software (Agilent technologies).

For measurements of metabolites in a group of 10 oocytes, three experiments using three independent female frogs were performed. Each experiment was conducted in five sample replicates per condition per frog; each sample consist of 10 pooled oocytes.

## Processing of samples for untargeted metabolomics at Metabolon (HD4 platform)

The samples were processed according to Metabolon's standard protocols at Metabolon Inc (Durham, NC, USA). The processes were conducted in an automated manner using the MicroLab STAR system from Hamilton. The samples were ultrasonicated in deionized water and a small portion of the homogenate was used to quantify the protein content with Bradford. The protein content was used for metabolomics data normalization.

The metabolite extraction process was conducted with recovery standards, which were added to the samples for the QC purposes. The samples were extracted with a series of organic and aqueous solvents under vigorous shaking for 2 min (Glen Mills GenoGrinder 2000). The samples were centrifuged, and the metabolite extract was divided into 4 aliquots, each dedicated for a different measurement strategy. The organic solvent was removed from the 4 sample aliquots using TurboVap (Zymark), and the samples were kept overnight under nitrogen flow.

## Untargeted metabolic measurements at Metabolon (HD4 platform)

To ensure the comprehensive metabolite coverage each of the four dried sample aliquots was reconstituted in solvents compatible with the given measurement method. The reconstitution solvents added to each sample contained a series of standards at fixed concentrations to ensure injection and chromatographic consistency. The reconstituted samples were analyzed in the following conditions: (1) Acidic positive ion (optimized for hydrophilic compounds). The extract was gradient eluted from a C18 column (Waters UPLC BEH C18–2.1×100 mm, 1.7 μm) with water and methanol containing 0.05% perfluoropentanoic acid and 0.1% formic acid; (2) Acidic positive ion (optimized for hydrophobic compounds). The extract was gradient eluted from C18 (Waters UPLC BEH C18–2.1×100 mm, 1.7 μm) with methanol, acetonitrile, water, 0.05% perfluoropentanoic acid, and 0.01% formic acid; (3) Basic negative ion. The extract was gradient eluted from a separate dedicated C18 column using methanol and water containing 6.5 mM ammonium bicarbonate at pH 8; and (4) Negative ionization. The extract was gradient eluted from a HILIC column (Waters UPLC BEH Amide 2.1×150 mm, 1.7 μm) using water and acetonitrile with 10 mM ammonium formate at pH 10.8.

For each of the methods the MS scan range varied, but covered 70–1000 m/z.

The untargeted metabolic profiling was conducted using Waters ACQUITY ultra-performance liquid chromatography (UPLC) and a Thermo Scientific Q-Exactive high resolution/accurate mass spectrometer interfaced with a heated electrospray ionization (HESI-II) source and Orbitrap mass analyzer, as previously described (PMID: 19624122).

The obtained raw data was extracted using Metabolon's hardware and software. The compounds were identified by comparison of peaks to library entries of purified standards using retention index, accurate mass match to the library ±10 ppm, and MS/MS forward and reverse scores between the experimental data and authentic standards as references. Manual assessment of library matches for each compound and each sample was conducted. The metabolic data was normalized to correct variations resulting from inter-day tuning differences in the instrument. Each compound was corrected in a run-day and normalized on protein content determined with Bradford.

## Processing of samples for lipidomics at Metabolon (CLP platform)

The samples for lipidomics measurements were also conducted at Metabolon Inc using Metabolon's standard protocols, performed in an automated manner by deploying the MicroLab STAR system from Hamilton. The samples were ultrasonicated in deionized water and a small portion of the homogenate was used to quantify the protein content with Bradford, for data normalization. Further sample processing was conducted in accordance with to modified Bligh-Dyer extraction (PMID: 13671378) using methanol/water/dichloromethane. The lipid extraction was conducted in the presence of internal standards. The organic extracts were dried under nitrogen flow using TurboVap (Zymark). The samples where reconstituted in dichloromethane: methanol (50:50) containing 10 mM ammonium acetate.

## Measurements of samples for lipidomics at Metabolon (CLP platform)

The reconstituted sample extracts were directly infused by deploying Shimadzu liquid chromatography (LC) with nano PEEK tubing as previously described (PMID: 29363076 and PMID: 22645248). The measurements were conducted in both positive and negative electrospray ionization modes using Sciex SelexIon-5500 QTRAP. The molecules were detected in multiple reaction monitoring (MRM) mode with a total of more than 1100 MRMs. The individual lipid species were quantified. It was achieved by calculation of the ratio of the signal intensity of each measured compound to the one of its assigned internal standards, followed by the multiplication of the concentration of internal standard added to the sample. The concentrations of the lipid class were calculated from the sum of all molecular species within a class. The fatty acid compositions were determined by calculating the proportion of each class comprised of individual fatty acids. The obtained data were normalized on protein content.

## Tandem LC MSMS for measurements of prostaglandins and S1P

For the measurements of prostaglandins and S1P, 10 x '10 oocytes' per condition were used; vehicle (30 min in 50 µl Ringer 1 X without P4), 5 min P4 (25 min in 50 µl Ringer 1 X without P4, then P4 was added for 5 min), and 30 min P4 (30 min in 50 µl Ringer 1 X containing P4 at $10^{-5}$ M). After the end of each treatment, oocytes were quickly washed with water and then flash-frozen. Three rounds of experiments from three different female frogs were performed and used for tandem LC-MS/MS measurement of arachidonic acid, S1P, and prostaglandins.

Eicosanoids were extracted as previously described (*Nithipatikom et al., 2003*). Briefly, pooled oocytes were vortexed in 1 ml solution containing 0.1% BHT, 800 µl ddH2O, 175 µl ethanol, and 25 µl acetic acid, followed by centrifugation at 1500 rpm for 3 min. The supernatant was loaded onto an ethanol:ddH2O (1:3) pre-conditioned C18 Bond Elut SPE column, washed with 20 ml ddH2O, then eluted with 5 ml ethyl acetate and collected in a glass tube. After the ethyl acetate layer was removed from the water layer from the bottom of the glass tubes, the water layer was extracted twice with ethyl acetate. All ethyl acetate portion was pooled, vacuum dried down and stored at –80 °C freezer. On the day of measurement, dried samples were reconstituted in 25 µl 35%Acetonitrile, 65% $H_2O$, and 0.1% acetic acid solution for LC/MS/MS data acquisition. For S1P measurement, pooled frog oocytes were extracted by 80% MeOH, supernatant from 80% MeOH extracts were dried down and reconstituted in 25 µl 35%Acetonitrile, 65% $H_2O$, and 0.1% acetic acid solution for LC/MS/MS data acquisition.

The tandem LC/MS/MS platform for prostaglandin and S1P measurement comprised of an Agilent Model 1290 Infinity II liquid chromatography system coupled to an Agilent 6460 Triple Quadrupole MS analyzer. An Agilent Zorbax SB-AQ reversed phase column (2.1×100 mm, 1.8 µm particle size), was used for the separation. Mobile phases consist of (A) 0.1% formic acid and 1 mM ammonium formate in 99% water 1% acetonitrile and (B) 0.1% formic acid and 1 mM ammonium formate in 99% acetonitrile H2O. Column temperature was set at 60 °C and autosampler temperature was at 4 °C. The flow rate was 0.4 mL/min. The following was applied: 0–2, 30% B; 2–12 min, to 65% B; 12–12.5 min, to 95%, 12.5–14.5 min, 95% B; 14.5–15 min, to 30% B; 15–20 min, 30% B. MRM transitions for qualitative and quantitative ions were acquired for PGA1, A2, B1, B2, D2/E2, I2. F1α, F2α. Note that PGD2 and PGE2 were not separatable by this method.

For the measurements of the eicosanoids, EETs, and HETEs – we used the services of Cayman chemicals. Single oocytes were incubated with either ethanol or P4 $10^{-5}$M in Ringer 1 x. 20 oocytes per condition were used. 30 min later, each oocyte was lysed within the tube by pipetting up and down, followed by spinning 3 x for 1000 g 10 min at 4 C. Supernatants were collected, flash-frozen and kept at –80 C prior to shipment/analysis.

## PLA2 assay

RNAs for mPR and ABHD2.S wt or ABHD2.S S/D/H mutants were used to generate recombinant proteins of mPR, ABHD2.S, and mPR/ABHD2.S, using the rabbit reticulocytes lysates Nuclease-treated kit and the transcend tRNA from Promega, as per the manufacturer's instruction. These different reactions were used to measure phospholipase A2 (PLA2) activity following the manufacturer's procedures (Abcam, Cambridge, UK), in the presence of ethanol or P4 ($10^{-5}$ M). No RNA control without substrate was used as background. PLA2 activity was corrected on reticulocytes without any RNA.

We used a coupled in vitro transcription/translation kit (ALiCECell-Free Protein Synthesis System, Sigma Aldrich) that allows the expression of difficult-to-produce proteins, such as membrane proteins, in microsomes using the pAlice02.His vector, which adds a melittin signal peptide to translocate the proteins into microsomes. To subclone mPRβ into the pALICE02 vector, pSGEM-mPR was PCR amplified using the following primers: 5' CTACCATGGCAATGACTACCGCAATCCTTGA-3' (Forward) 5'-CTAGGTACCTCAGTGATGGTGATGGTGATGAAGTTCTTTTCTGGCCAACT-3' (Reverse). The resulting PCR product was cut with NcoI and KpnI, gel purified, and ligated into NcoI/KpnI of pALICE02. To subclone ABHD2 in pALICE02, pSGEM-ABHD2 was PCR amplified using the following primers: 5'-ACTGATATCATGGATGCGATAGTGGAAACCC-3'(Forward) and 5'-CTAGGTACCTCAGTGATGGT GATGGTGATGTTTATGGTCAGACTCAGCAGCC-3'(Reverse). The resulting PCR product was cut with EcoRV and KpnI, gel purified, and ligated into pALICE02 cut with NcoI and Klenow treated to produce a blunt end then cut with KpnI. All constructs were verified by DNA sequencing and by analytical endonuclease restriction enzyme digestion. The pAlice02 vector allows for the expression of C-terminally His-tagged mPRβ and ABHD2. Briefly, DNA was added to the pALiCE lysates, followed by incubation at 25 °C for 48 hr with constant shaking at 700 rpm. The expression of ABHD2 and mPRβ were confirmed by western blot, before proceeding to the PLA2 assay.

## Statistics

Data are presented as mean ± SEM. Each set of experiments was at least repeated three times. Groups were compared using the Prism 9 software (GraphPad) using the statistical tests indicated in the figure legend. Statistical significance is indicated by p-values (ns, not significant; ***$p \leq 0.001$; **$p \leq 0.01$; *$p \leq 0.05$).

## Materials availability statement

All clones or reagents generated in the course of this study are readily available to colleagues in the scientific community without any restrictions. If interested please contact the corresponding author, Khaled Machaca at khm2002@qatar-med.cornell.edu.

## Acknowledgements

We are grateful to the Bioinformatics and Virtual Metabolomics Core at Weill Cornell Medicine Qatar for their support in the metabolomics and lipidomic studies. We also thank the Vivarium and Microscopy Cores at WCMQ for their support in multiple experiments. This work as well as the Cores are supported by the Biomedical Research Program at Weill Cornell Medical College in Qatar (BMRP), a program funded by the Qatar Foundation. Additional funding was provided by NPRP-Standard (NPRP-S) 13th Cycle grant 13 S-0206-200274 from the Qatar National Research Fund (a member of Qatar Foundation). The findings herein reflect the work and are solely the responsibility of the authors. This work was also supported by NIH R01AR076029 and NIH R21ES032347 to QC.

## Additional information

### Funding

| Funder | Grant reference number | Author |
| --- | --- | --- |
| Qatar Foundation | BMRP | Khaled Machaca |
| Qatar National Research Fund | NPRP13S-0206-200274 | Nancy Nader Khaled Machaca |
| NIH Office of the Director | R01AR076029 | Qiuying Chen |
| NIH Blueprint for Neuroscience Research | R21ES032347 | Qiuying Chen |

The funders had no role in study design, data collection and interpretation, or the decision to submit the work for publication.

## Author contributions
Nancy Nader, Conceptualization, Data curation, Formal analysis, Investigation, Methodology, Writing – original draft, Writing – review and editing; Lama Assaf, Lubna Zarif, Sharan Yadav, Maya Dib, Investigation, Methodology; Anna Halama, Formal analysis, Investigation; Nabeel Attarwala, Investigation; Qiuying Chen, Formal analysis, Investigation, Methodology; Karsten Suhre, Steven Gross, Formal analysis; Khaled Machaca, Conceptualization, Data curation, Formal analysis, Supervision, Funding acquisition, Writing – original draft, Project administration, Writing – review and editing

## Author ORCIDs
Nancy Nader http://orcid.org/0000-0003-1688-8174
Qiuying Chen https://orcid.org/0000-0001-5909-3959
Khaled Machaca https://orcid.org/0000-0001-6215-2411

## Ethics
This study was performed in accordance with the recommendations in the Guide for the Care and Use of Laboratory Animals of the National Institutes of Health. All of the animals were handled according to approved institutional animal care and use committee (IACUC) protocols (#2011-0035) of the Weill Cornell Medicine Qatar. The protocol was approved by the IACUC committee of Weill Cornell Medicine Qatar.

Reviewer #1 (Public review): https://doi.org/10.7554/eLife.92635.3.sa1
Reviewer #2 (Public review): https://doi.org/10.7554/eLife.92635.3.sa2
Reviewer #3 (Public review): https://doi.org/10.7554/eLife.92635.3.sa3
Author response https://doi.org/10.7554/eLife.92635.3.sa4

# Additional files

## Supplementary files
Supplementary file 1. List of reagents. (a) List of antibodies, chemicals, and reagents. (b) List of antisense oligonucleotides and primers.

MDAR checklist

Source data 1. Full Western blots.

Source data 2. Labelled full Western Blots.

Source data 3. Means and p-values for ratio fold changes for lipids analyzed on the CLP platform at 5 min time point after treatment with progesterone (P4). Means and p-values for ratio fold changes for lipids analyzed on the CLP platform at 30 min time point after treatment with P4. Means and p-values for ratio fold changes for lipids analyzed on the HD4 platform at 5 min time point after treatment with P4. Means and p-values for ratio fold changes for other metabolites analyzed on the HD4 platform at 5 min time point after treatment with P4. Means and p-values for ratio fold changes for lipids analyzed on the HD4 platform at 30 min time point after treatment with P4. Means and p-values for ratio fold changes for other metabolites analyzed on the HD4 platform at 30 min time point after treatment with P4. Means and p-values for ratio fold changes for different compounds at 30 min time point after treatment with P4 in single oocyte metabolomics.

## Data availability
All data generated or analysed during this study are included in the manuscript and associated source data files. The metabolomics datasets in the source data tables include the metabolomics and lipidomics platforms used as explained in details in the methods section.

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
