## [Editor Report · eLife Assessment]

This **important** study provides **solid** evidence for a non-genomic action of progesterone in *Xenopus* oocyte activation. The findings demonstrate that two non-genomic progesterone receptors, ABHD2 and mPRb, function as a novel progesterone-stimulated phospholipase A2. The findings will be of broad interest to reproductive endocrinologists and physiologists.

---

## [Referee Report · Reviewer #1 (Public review)]

Summary:

Numerous pathways have been proposed to elucidate the nongenomic actions of progesterone within both male and female reproductive tissues. The authors employed the *Xenopus* oocyte system to investigate the PLA2 activity of ABHD2 and the downstream lipid mediators in conjunction with mPRb and P4, on their significance in meiosis. The research has been conducted extensively and is presented clearly.

Strengths:

While the interaction between membranous PR and ABHD2 is not a novel concept, this present study exhibits several strengths:

(1) mPRbeta, a member of the PAQR family, has been elusive in terms of detailed signal transduction. Through mutation studies involving the Zn binding domain, the authors discovered that the hydrolase activity of mPRbeta is not essential for meiosis and oocyte maturation. Instead, they suggest that ABHD2, acting as a coreceptor of mPRbeta, demonstrates phospholipase activity, indicating that downstream lipid mediators may play a dominant role when stimulated by progesterone.

(2) Extensive exploration of downstream signaling pathways and the identification of several potential meiotic activity-related lipid mediators make this aspect of the study novel and potentially significant.

Weaknesses:

However, there are some weaknesses and areas that need further clarification:

(1) The mechanism governing the molecular assembly of mPRbeta and ABHD2 remains unclear. Are they constitutively associated or is their association ligand-dependent? Does P4 bind not only to mPRbeta but also to ABHD2, as indicated in Figure 6J? In the latter case, the reviewer suggests that the authors conduct a binding experiment using labeled P4 with ABHD2 to confirm this interaction and assess any potential positive or negative cooperativity with a partner receptor.

(2) The authors have diligently determined the metabolite profile using numerous egg cells. However, the interpretation of the results appears incomplete, and inconsistencies were noted between Figure 2F and Supplementary Figure 2C. Furthermore, PGE2 and D2 serve distinct roles and have different elution patterns by LC-MS/MS, thus requiring separate measurements. In addition, the extremely short half-life of PGI2 necessitates the measurement of its stable metabolite, 6-keto-PGF1a, instead. The authors also need to clarify why they measured PGF1a but not PGF2a. Unfortunately, even in the revision, authors did not adequately address the last issue (differential measurements of PGD2 and E2, 6-keto-PG!alpha be determined instead of PGI2).

(3) Although they propose PGs, LPA and S1P are important downstream mediators, the exact roles of the identified lipid mediators have not been clearly demonstrated, as receptor expression and activation were not demonstrated. While the authors showed S1PR3 expression and its importance by genetic manipulation, there was no observed change in S1P levels following P4 treatment (Supplementary Figure 2D). It is essential to identify which receptors (subtypes) are expressed and how downstream signaling pathways (PKA, Ca, MAPK, etc.) relate to oocyte phenotypes.

These clarifications and further experiments would enhance the overall impact and comprehensiveness of the study.

Comments on revisions:

Need correction and addition for differential analyses of PGD2 and PGE2, and measurement of 6-keto-PGF1alpha instead of PGI2 (Figure 2F). PGI2 is extremely unstable (T1/2, 1 min in neutral buffer) and rapidly converted nonenzymically to 6-keto-PGF1a.

---

## [Referee Report · Reviewer #2 (Public review)]

Summary:

This interesting paper examines the earliest steps in progesterone-induced frog oocyte maturation, an example of non-genomic steroid hormone signaling that has been studied for decades but is still very incompletely understood. In fish and frog oocytes it seems clear that mPR proteins are involved, but exactly how they relay signals is less clear. In human sperm, the lipid hydrolase ABHD2 has been identified as a receptor for progesterone, and so the authors here examine whether ABHD2 might contribute to progesterone-induced oocyte maturation as well. The main results are:

(1) Knocking down ABHD2 makes oocytes less responsive to progesterone, and ectopically expressing ABHD2.S (but not the shorter ABHD2.L gene product) partially rescues responsiveness. The rescue depends upon the presence of critical residues in the protein's conserved lipid hydrolase domain, but not upon the presence of critical residues in its acyltransferase domain.

(2) Treatment of oocytes with progesterone causes a decrease in sphingolipid and glycerophospholipid content within 5 min. This is accompanied by an increase in LPA content and arachidonic acid metabolites. These species may contribute to signaling through GPCRs. Perhaps surprisingly, there was no detectable increase in sphingosine-1-phosphate, which might have been expected given the apparent substantial hydrolysis of sphingolipids. The authors speculate that S1P is formed and contributes to signaling but diffuses away.

(3) Pharmacological inhibitors of lipid-metabolizing enzymes support, for the most part, the inferences from the lipidomics studies, although there are some puzzling findings. The puzzling findings may be due to uncertainty about whether the inhbitors are working as advertised.

(4) Pharmacological inhibitors of G-protein signaling support a role for G-proteins and GPCRs in this signaling, although again there are some puzzling findings.

(5) Reticulocyte expression supports the idea that mPRβ and ABHD2 function together to generate a progesterone-regulated PLA2 activity.

(6) Knocking down or inhibiting ABHD2 inhibited progesterone-induced mPRβ internalization, and knocking down ABHD2 inhibited SNAP25∆20-induced maturation.

Strengths:

All in all, this could be a very interesting paper and a nice contribution. The data add a lot to our understanding of the process, and, given how ubiquitous mPR and AdipoQ receptor signaling appear to be, something like this may be happening in many other physiological contexts.

Weaknesses:

I have several suggestions for how to make the main points more convincing.

Main criticisms:

(1) The ABHD2 knockdown and rescue, presented in Fig 1, is one of the most important findings. It can and should be presented in more detail to allow the reader to understand the experiments better. E.g.: the antisense oligos hybridize to both ABHD2.S and ABHD2.L, and they knock down both (ectopically expressed) proteins. Do they hybridize to either or both of the rescue constructs? If so, wouldn't you expect that both rescue constructs would rescue the phenotype, since they both should sequester the AS oligo? Maybe I'm missing something here.

In addition, it is critical to know whether the partial rescue (Fig 1E, I, and K) is accomplished by expressing reasonable levels of the ABHD2 protein, or only by greatly overexpressing the protein. The author's antibodies do not appear to be sensitive enough to detect the endogenous levels of ABHD2.S or .L, but they do detect the overexpressed proteins (Fig 1D). The authors could thus start by microinjecting enough of the rescue mRNAs to get detectable protein levels, and then titer down, assessing how low one can go and still get rescue. And/or compare the mRNA levels achieved with the rescue construct to the endogenous mRNAs.

Finally, please make it clear what is meant by n = 7 or n = 3 for these experiments. Does n = 7 mean 7 independently lysed oocytes from the same frog? Or 7 groups of, say, 10 oocytes from the same frog? Or different frogs on different days? I could not tell from the figure legends, the methods, or the supplementary methods. Ideally one wants to be sure that the knockdown and rescue can be demonstrated in different batches of oocytes, and that the experimental variability is substantially smaller than the effect size.

(2) The lipidomics results should be presented more clearly. First, please drop the heat map presentations (Fig 2A-C) and instead show individual time course results, like those shown in Fig 2E, which make it easy to see the magnitude of the change and the experiment-to-experiment variability. As it stands, the lipidomics data really cannot be critically assessed.

[Even as heat map data go, panels A-C are hard to understand. The labels are too small, especially on the heat map on the right side of panel B. And the 25 rows in panel C are not defined (the legend makes me think the panel is data from 10 individual oocytes, so are the 25 rows 25 metabolites? If so, are the individual oocyte data being collapsed into an average? Doesn't that defeat the purpose of assessing individual oocytes?) And those readers with red-green colorblindness (8% of men) will not be able to tell an increase from a decrease. But please don't bother improving the heat maps; they should just be replaced with more-informative bar graphs or scatter plots.]

(3) The reticulocyte lysate co-expression data are quite important, and are both intriguing and puzzling. My impression had been that to express functional membrane proteins, one needed to add some membrane source, like microsomes, to the standard kits. Yet it seems like co-expression of mPR and ABHD2 proteins in a standard kit is sufficient to yield progesterone-regulated PLA2 activity. I could be wrong here-I'm not a protein expression expert-but I was surprised by this result, and I think it is critical that the authors make absolutely certain that it is correct. Do you get much greater activities if microsomes are added? Are the specific activities of the putative mPR-ABHD2 complexes reasonable?

Comments on revisions:

The authors have satisfied my concerns with their response letter and revisions.

---

## [Referee Report · Reviewer #3 (Public review)]

Summary:

The authors report two P4 receptors, ABHD2 and mPRβ that function as co-receptors to induce PLA2 activity and thus drive meiosis. In their experimental studies, the authors knock down ABHD2 and demonstrated inhibition of oocyte maturation and inactivation of Plk1, MAPK, and MPF, which indicated that ABHD2 is required for P4-induced oocyte maturation. Next, they showed three residues (S207, D345, H376) in the lipase domain that are crucial for ABHD2 P4-mediated oocyte maturation in functional assays. They performed global lipidomics analysis on mPRβ or ABHD2 knockdown oocytes, among which the downregulation of GPL and sphingolipid species were observed and enrichment in LPA was also detected using their metabolomics method. Furthermore, they investigated pharmacological profiles of enzymes predicted to be important for maturation based on their metabolomic analyses and ascertained the central role for PLA2 in inducing oocyte maturation downstream of P4. They showed the modulation of S1P/S1PR3 pathway on oocyte maturation and potential role for or Gαs signaling and potentially Gβγ downstream of P4.

Strengths:

The authors make a very interesting finding that ABHD2 has PLA2 catalytic activity but only in the presence of mPRβ and P4. Finally, they provided supporting data for a relationship between ABHD2/PLA2 activity and mPRβ endocytosis and further downstream signaling. Collectively, this research report defines early steps in nongenomic P4 signaling, which is of broad physiological implications.

Weaknesses:

There were concerns with the pharmacological studies presented. Many of these inhibitors are used at high (double digit micromolar) concentrations that could result in non-specific pharmacological effects and the authors have provided very little data in support of target engagement and selectivity under the multiple experimental paradigms. In addition, the use of an available ABHD2 small molecule inhibitor was lacking in these studies.

Comments on revisions:

In the revised manuscript, the authors have addressed my major concerns.

---

## [Author Response]

The following is the authors’ response to the original reviews.

**eLife Assessment:**
“…However, the findings are reliant on high concentrations of inhibitor drugs, and mechanistic details about the molecular interaction and respective functions of ABHD2 and mPRb are incomplete.”

As discussed below in the response to Reviewers the drug concentrations used span the full dose response of the active range of each drug. In cases where the drug concentrations required to block oocyte maturation where significantly higher than those reported in the literature, we considered those drugs ineffective. In terms of the molecular details of the mechanistic interaction between mPRb and ABHD2, we now provide additional data confirming their molecular interaction to produce PLA2 activity where each protein alone is insufficient. Although these new studies provide more mechanistic insights, there remains details of the ABHD2-mPR interactions that would need to be addressed in future studies which are beyond the scope of the current already extensive study.

**Public Reviews:**

**Reviewer 1**
(1) The mechanism governing the molecular assembly of mPRbeta and ABHD2 remains unclear. Are they constitutively associated or is their association ligand-dependent? Does P4 bind not only to mPRbeta but also to ABHD2, as indicated in Figure 6J? In the latter case, the reviewer suggests that the authors conduct a binding experiment using labeled P4 with ABHD2 to confirm this interaction and assess any potential positive or negative cooperativity with a partner receptor.

The co-IP experiments presented in Figure 5E argue that the two receptors are constitutively associated at rest before exposure to P4; but at low levels since addition of P4 increases the association between mPRβ and ABHD2 by ~2 folds. Importantly, we know from previous work (Nader et al., 2020) and from imaging experiments in this study that mPR recycles in immature oocytes between the PM and the endosomal compartment. It is not clear at this point within which subcellular compartment the basal association of mPR and ABHD2 occurs. We have tried to elucidate this point but have not been able to generate a functional tagged ABHD2. We generated GFP-tagged ABHD2 at both the N- and C-terminus but these constructs where not functional in terms of their ability to rescue ABHD2 knockdown. This prevented us from testing the association dynamics between ABHD2 and mPR.

Regarding whether ABHD2 in the oocyte directly binds P4 or not, we had in the initial submission no data directly supporting this rather we based the cartoon in Fig. 6J on the findings from Miller et al. (Science 2016) who showed that ABHD2 in sperm binds biotinylated P4. With the use of a new expression system to produce ABHD2 in vitro (please see below) we were able to try the experiment suggested by the Reviewer. In vitro expressed ABHD2 was incubated with biotinylated P4, and binding tested on a streptavidin column. Under these conditions we could not detect any specific binding of P4 to ABHD2, however, these experiments remain somewhat preliminary and would require validation using additional approaches to conclusively test whether *Xenopus* ABHD2 binds P4 or not. The discrepancy with the Miller et al. findings could be species specific as they tested mammalian ABHD2.

(2) The authors have diligently determined the metabolite profile using numerous egg cells. However, the interpretation of the results appears incomplete, and inconsistencies were noted between Figure 2B and Supplementary Figure 2C. Furthermore, PGE2 and D2 serve distinct roles and have different elution patterns by LC-MS/MS, thus requiring separate measurements. In addition, the extremely short half-life of PGI2 necessitates the measurement of its stable metabolite, 6-keto-PGF1a, instead. The authors also need to clarify why they measured PGF1a but not PGF2a.

We believe the Reviewer meant to indicate discrepancies between Fig. 2E (not 2B) and Supp. Fig. 2C. Indeed, the Reviewer is correct, and this is because Fig. 2E shows pooled normalized data on a per PG species and frog, whereas Supp. Fig. 2E shows and example of absolute raw levels from a single frog to illustrate the relative basal abundance of the different PG species. We had failed to clarify this in the Supp. Fig. 2E figure legend, which we have now added in the revised manuscript. So, the discrepancies are due to variation between different donor animals which is highlighted in Supp. Fig. 2A. Furthermore, to minimize confusion, in the revised manuscript we revised Supp. Fig. 2C to show only PG levels at rest, to illustrate basal levels of the different PG species relative to each other, which is the goal of this supplemental figure.

(3) Although they propose PGs, LPA, and S1P are important downstream mediators, the exact roles of the identified lipid mediators have not been clearly demonstrated, as receptor expression and activation were not demonstrated. While the authors showed S1PR3 expression and its importance by genetic manipulation, there was no observed change in S1P levels following P4 treatment (Supplementary Figure 2D). It is essential to identify which receptors (subtypes) are expressed and how downstream signaling pathways (PKA, Ca, MAPK, etc.) relate to oocyte phenotypes.

We agree conceptually with the Reviewer that identifying the details of the signaling of the different GPCRs involved in oocyte maturation would be interesting. However, our lipidomic data argue that the activation of a PLA2 early in the maturation process in response to P4 leads to the production of multiple lipid messengers that would activate GPCRs and branch out the signaling pathway to activate various pathways required for the proper and timely progression of oocyte maturation. Preparing the egg for fertilization is complex; so, it is not surprising that a variety of pathways are activated simultaneously to properly initiate both cytoplasmic and nuclear maturation to transition the egg from its meiotic arrest state to be ready to support the rapid growth during early embryogenesis. We focus on the S1P signaling pathway specifically because, as pointed out by the Reviewer, we could not detect an increase in S1P even though our metabolomic data collectively argued for an increase. Our results on the S1P pathway -as well as a plethora of other studies historically in the literature that we allude to in the manuscript- argue that these different GPCRs support and regulate oocyte maturation, but they are not essential for the early maturation signaling pathway. For example, for S1P, as shown in Figure 4, the delay/inhibition of oocyte maturation due to S1PR3 knockdown can be reversed at high levels of P4, which presumably leads to higher levels of other lipid mediators that would bypass the need for signaling through S1PR3. This is reminiscent of the kinase cascade driving oocyte maturation where there is significant redundancy and feedback regulation. Therefore, analyzing each receptor subtype that may regulate the different PG species, LPA, and S1P would be a tedious and time-consuming undertaking that goes beyond the scope of the current manuscript. More importantly based on the above arguments, we suggest that findings from such an analysis, similar to the conclusions from the S1PR3 studies (Fig. 4), would show a modulatory role on oocyte maturation rather than a core requirement for the maturation process as observed with mPR and ABHD2. Thus they would provide relatively little insights into the core signaling pathway driving P4-mediated oocyte maturation.

**Reviewer 2:**
(1) The ABHD2 knockdown and rescue, presented in Fig 1, is one of the most important findings. It can and should be presented in more detail to allow the reader to understand the experiments better. E.g.: the antisense oligos hybridize to both ABHD2.S and ABHD2.L, and they knock down both (ectopically expressed) proteins. Do they hybridize to either or both of the rescue constructs? If so, wouldn't you expect that both rescue constructs would rescue the phenotype since they both should sequester the AS oligo? Maybe I'm missing something here.

For the ABHD2 rescue experiment, the ABHD2 constructs (S or L) were expressed 48 hrs before the antisense was injected. The experiment was conducted in this way to avoid the potential confounding issue of both constructs sequestering the antisense. The assumption is that the injected RNA after protein expression would be degraded thus allowing the injected antisense to target endogenous ABHD2. The idea is to confirm that ABHD2.S expression alone is sufficient to rescue the antisense knockdown as confirmed experimentally.

However, to further confirm the rescue, we performed the experiment in a different chronological order, where we started with injecting the antisense to knock down endogenous ABHD2 and this was followed 24 hrs later by expressing wild type ABHD2.S. As shown in Author response image 1 this also rescues the knockdown.

**Author response image 1. sa4fig1:** ABHD2 knockdown and rescue. Oocytes were injected with control antisense (Ctrl AS) or specific ABHD2 antisense (AS) oligonucleotides and incubated at 18 oC for 24 hr. Oocytes were then injected with mRNA to overexpress ABHD.S for 48 hr and then treated with P4 overnight. The histogram shows % GVBD in naïve, oocytes injected with control or ABHD2 antisense with or without mRNA to overexpress ABHD2.S.

In addition, it is critical to know whether the partial rescue (Fig 1E, I, and K) is accomplished by expressing reasonable levels of the ABHD2 protein, or only by greatly overexpressing the protein. The author's antibodies do not appear to be sensitive enough to detect the endogenous levels of ABHD2.S or .L, but they do detect the overexpressed proteins (Fig 1D). The authors could thus start by microinjecting enough of the rescue mRNAs to get detectable protein levels, and then titer down, assessing how low one can go and still get rescue. And/or compare the mRNA levels achieved with the rescue construct to the endogenous mRNAs.

The dose response of ABHD2 protein expression in correlation with rescue of the ABHD2 knockdown is shown indirectly in Figure 1I and 1J. In experiments ABHD2 knockdown was rescued using either the WT protein or two mutants (H120A and N125A). All three constructs rescued ABHD2 KD with equal efficiency (Fig. 1I), eventhough their expression levels varied (Fig. 1J). The WT protein was expressed at significantly higher levels than both mutants, and N125A was expressed at higher levels than H120A (Fig. 1J), note the similar tubulin loading control. Crude estimation of the WBs argues for the WT protein expression being ~3x that of H120A and ~2x that of N125A, yet all three have similar rescue of the ABHD2 knockdown (Fig. 1I). This argues that low levels of ABHD2 expression is sufficient to rescue the knockdown, consistent with the catalytic enzymatic nature of the ABHD2 PLA2 activity.

Finally, please make it clear what is meant by n = 7 or n = 3 for these experiments. Does n = 7 mean 7 independently lysed oocytes from the same frog? Or 7 groups of, say, 10 oocytes from the same frog? Or different frogs on different days? I could not tell from the figure legends, the methods, or the supplementary methods. Ideally one wants to be sure that the knockdown and rescue can be demonstrated in different batches of oocytes, and that the experimental variability is substantially smaller than the effect size.

The n reflects the number of independent female frogs. We have added this information to the figure legends. For each donor frog at each time point 10-30 oocytes were used.

(2) The lipidomics results should be presented more clearly. First, please drop the heat map presentations (Fig 2A-C) and instead show individual time course results, like those shown in Fig 2E, which make it easy to see the magnitude of the change and the experiment-to-experiment variability. As it stands, the lipidomics data really cannot be critically assessed.[Even as heat map data go, panels A-C are hard to understand. The labels are too small, especially on the heat map on the right side of panel B. The 25 rows in panel C are not defined (the legend makes me think the panel is data from 10 individual oocytes, so are the 25 rows 25 metabolites? If so, are the individual oocyte data being collapsed into an average? Doesn't that defeat the purpose of assessing individual oocytes?) And those readers with red-green colorblindness (8% of men) will not be able to tell an increase from a decrease. But please don't bother improving the heat maps; they should just be replaced with more informative bar graphs or scatter plots.]

We have revised the lipidomics data as requested by the Reviewer. The Reviewer asked that we show the data as a time course with each individual frog as in Fig. 2E. This turns out to be confusing and not a good way to present the data (please see Author response image 2).

**Author response image 2. sa4fig2:** Metabolite levels from 5 replicates of 10 oocytes each at each time point were measured and averaged per frog and per time point. Fold change was measured as the ratio at the 5- and 30-min time points relative to untreated oocytes (T0). FCs that are not statistically significant are shown as faded. Oocytes with mPR knockdown (KD) are boxed in green and ABHD2-KD in purple.

We therefore revised the metabolomics data as follow to improve clarity. The changes in the glycerophospholipids and sphingolipids determined on the Metabolon CLP platform (specific for lipids) are now shown as single metabolites clustered at the levels of species and pathways and arranged for the 5- and 30-min time points sequentially on the same heatmap as requested (Fig. 2B). This allows for a quick visual overview of the data that clearly shows the decrease in the lipid species following P4 treatment in the control oocytes and not in the mPR-KD or ABHD2-KD cells (Fig. 2B). The individual species are listed in Supplemental Tables 1 and 2. We also revised the Supplemental Tables to include the values for the non-significant changes, which were omitted from the previous submission.

We revised the metabolomics data from the HD4 platform in a similar fashion but because the lipid data were complimentary and less extensive than those from the CLP platform, we moved that heatmap to Supplemental Fig. 2B.

For the single oocyte metabolomics, we now show the data as the correlation between FC and p value, which clearly shows the upregulated (including LPA) and downregulated metabolites at T30 relative to T0 (Fig. 2C). The raw data is now shown in a new Supplemental Table 7.

(3) The reticulocyte lysate co-expression data are quite important and are both intriguing and puzzling. My impression had been that to express functional membrane proteins, one needed to add some membrane source, like microsomes, to the standard kits. Yet it seems like co-expression of mPR and ABHD2 proteins in a standard kit is sufficient to yield progesterone-regulated PLA2 activity. I could be wrong here - I'm not a protein expression expert - but I was surprised by this result, and I think it is critical that the authors make absolutely certain that it is correct. Do you get much greater activities if microsomes are added? Are the specific activities of the putative mPR-ABHD2 complexes reasonable?

We thank the Reviewer for this insightful comment. We agree that this is a critical result that would benefit from cross validation, especially given the low level of PLA2 activity detected in the reticulocyte lysate expression system. We have therefore expanded these studies using another in vitro expression system with microsomal membranes based on tobacco extracts (ALiCECell-Free Protein Synthesis System, Sigma Aldrich) to enhance production and stability of the expressed receptors as suggested by the Reviewer. We further prepared virus-like particles (VLPs) from cells expressing each receptor individually or both receptors together. We however could not detect any PLA2 activity from the VLPs. We thus focused on the coupled in vitro transcription/translation tobacco extracts that allow the expression of difficult-to-produce membrane proteins in microsomes. This kit targets membrane protein directly to microsomes using a microsome targeting melittin signal peptide. This system took significant time and effort to troubleshoot and adapt to mPR and ABHD2 expression. We were however ultimately able to produce significantly higher amounts of both ABHD2 and mPRb, which were readily detected by WBs (Supplemental Fig. 4I). In contrast, we could not reliably detect mPR or ABHD2 using WBs from reticulocyte lysates given the limited amounts produced.

Similarly to our previous findings with proteins produced in reticulocytes, expression of ABHD2 or mPRβ alone was not associated with an increase in PLA2 activity over a two-hour incubation period (Fig. 5C). It is worth noting here that the tobacco lysates had high endogenous PLA2 activity. However, co-expression of both mPRb and ABHD2 produced robust PLA2 activity that was significantly higher than that detected in reticulocyte lysate system (Fig. 5C). Surprisingly, however this PLA2 activity was P4 independent as it was observed when both receptors are co-expressed in the absence of P4.

These results validate our earlier conclusion that PLA2 activity requires both mPR and ABHD2, so their interaction in needed for enzymatic activity. It is interesting however that in the tobacco expression system this mPR-ABHD2 PLA2 activity becomes for the most part P4 independent. As the tobacco expression system forces both ABHD2 and mPR into microsomes using a signal sequence, the two receptors are enriched in the same vesicular compartment. As they can interact independently of P4 as shown in the co-IP experiments in immature oocytes (Fig. 5D), their forced co-expression in the same microsomal compartment could lead to their association and thus PLA2 activity. This is an attractive possibility that fits the current data, but would need independent validation.

**Reviewer 3:**

*There were concerns with the pharmacological studies presented. Many of these inhibitors are used at high (double-digit micromolar) concentrations that could result in non-specific pharmacological effects and the authors have provided very little data in support of target engagement and selectivity under the multiple experimental paradigms. In addition, the use of an available ABHD2 small molecule inhibitor was lacking in these studies.*

For the inhibitors used we performed a full dose response to define the active concentrations. So, inhibitors were not used at one high dose. We then compared the EC50 for each active inhibitor to the reported EC50 in the literature (Table 1). The inhibitors were deemed effective only if they inhibited oocyte maturation within the range reported in the literature. This despite the fact that frog oocytes are notorious in requiring higher concentrations of drug given their high lipophilic yolk content, which acts as a sponge for drugs. So our criteria for an effective inhibitor are rather stringent.

Based on these criteria, only 3 inhibitors were ‘effective’ in inhibiting oocyte maturation: Ibuprofen, ACA and MP-A08 with relative IC50s to those reported in the literature of 0.7, 1.1, and 1.6 respectively. Ibuprofen targets Cox enzymes, which produce prostaglandins. We independently confirmed an increase in PGs in response to P4 in oocytes thus validating the drug inhibitory effect. ACA blocks PLA2 and inhibits maturation, a role supported by the metabolomics analyses that shows decrease in the PE/PE/LPE/LPC species; and by the ABHD2-mPR PLA2 activity following in vitro expression. Finally, MP-A08 blocks sphingosine kinase activity, which role is supported by the metabolomics showing a decrease in sphingosine levels in response to P4; and our functional studies validating a role for the S1P receptor 3 in oocyte maturation.

As pointed out by the Reviewer, other inhibitors did block maturation at very high concentration, but we do not consider these as effective and have not implicated the blocked enzymes in the early steps of oocyte maturation. To clarify this point, we edited the summary panel (now Fig. 2D) to simplify it and highlight the inhibitors with an effect in the reported range in red and those that don’t inhibit based on the above criteria in grey. Those with intermediate effects are shown in pink. We hope these edits clarify the inhibitors studies.

**Recommendations For the Authors**

**Reviewer 2:**
(1) Introduction, para 1. Please change "mPRs mediated" to "mPR-mediated".

Done

(2) Introduction, para 2. Please change "cyclin b" to "cyclin B".

Done

(3) Introduction, para 2. Please change "that serves" to "which serves".

Done

(4) Introduction, para 4. I know that the authors have published evidence that "a global decrease in cAMP levels is not detectable" (2016), but old work from Maller and Krebs (JBC 1979) did see an early, transient decrease after P4 treatment, and subsequent work from Maller said that there was both a decrease in adenylyl cyclase activity and an increase in cAMP activity. Perhaps it would be better to say something like "early work showed a transitory drop in cAMP activity within 1 min of P4 treatment (Maller), although later studies failed to detect this drop and showed that P4-dependent maturation proceeds even when cAMP is high (25)".

We agree and thank the Reviewer for this recommendation. The text was revised accordingly.

(5) Results, para 1. Based on the results in Fig 1B, one should probably not assert that ABHD2 is expressed "at levels similar to those of mPRβ in the oocyte"-with different mRNAs and different PCR primers, it's hard to say whether they are similar or not. The RNAseq data from Xenbase in Supp Fig 1 supports the idea that the ABHD2 and mPRβ mRNAs are expressed at similar levels at the message level, although of course mRNA levels and protein levels do not correlate well when different gene products are compared (Wuhr's 2014 Curr Biol paper reported correlation coefficients of about 0.3).

We agree and have changed the text as follow to specifically point out to RNA: “we confirmed that ABHD2 RNA is expressed in the oocyte at levels similar to those of mPRβ RNA (Fig. 1B).”

(6) Results, para 2. It would be worth pointing out that since an 18 h incubation with microinjected antisense oligos was sufficient to substantially knock down both the ABHD2 mRNAs (Fig 1C) and the ectopically-expressed proteins (Fig 1D), the mRNA and protein half-lives must be fairly short, on the order of a few hours or less.

Done

(7) Figure 1. Please make the western blots (especially Fig 1D) and their labeling larger. These are key results and as it stands the labeling is virtually unreadable on printed copies of the figures. I'm not sure about eLife's policy, but many journals want the text in figures to be no smaller than 5-7 points at 100% size.Likewise for many of the western blots in subsequent figures.

As requested by the Reviewer we have increased the font and size of all Western blots in the Figures.

(8) Figure 1E, G. I am not sure one should compare the effectiveness of the ABHD2 rescue (Fig 1E) and the mPRβ rescue (Fig 1G). Even if these were oocytes from the same frog, we do not know how the levels of the overexpressed ABHD2 and mPRβ proteins compare. E.g. maybe ABHD2 was highly overexpressed and mPRβ was overexpressed by a tiny amount.

Although this is a possibility, the expression levels of the proteins here is not of much concern because we previously showed that mPRβ expression effectively rescues mPRβ antisense knockdown which inhibits maturation (please see (Nader et al., 2020)). This argues that at the levels of mRNA injected mPR is functional to support maturation, yet it does not rescue ABHD2 knockdown to the same levels (Fig. 1G). With that it is fair to argue that mPRβ is not as effective at rescuing ABHD2 KD maturation.

(9) Inhibitor studies: There are two likely problems in comparing the observed potencies with legacy data - in vitro vs in vivo data and frog vs. mammalian data. Please make it clear what is being compared to what when you are comparing legacy data.

The legacy data are from the literature based on the early studies that defined the IC50 for inhibition primarily using in vivo models (cell line mostly) but not oocytes. Typically, frog oocytes require significantly higher concentrations of inhibitors to mediate their effect because of the high lipophilic yolk content which acts as a sponge for some drugs. So, the fact that the drugs that are effective in inhibiting oocyte maturation (ACA, MP-A08, and Ibuprofen) work in a similar or lower concentration range to the published IC<sub50 gives us confidence as to the specificity of their effect. We have revised Table 1 to include the reference for each IC<sub50 value from the literature to allow the reader to judge the exact model and context used.

(10) Isn't it surprising that Gas seems to promote maturation, given the Maller data (and data from others) that cAMP and PKA oppose maturation (see also the authors' own Fig 1A) and the authors' previous data sees no positive effect (minor point 7 above)?

We show that a specific Gas inhibitor NF-449 inhibits maturation (although at relatively high concentrations), which is consistent with a positive role for Gas in oocyte maturation. We argue based on the lipidomics data and the inhibitors data that GPCRs play a modulatory role and not a central early signaling role in terms of releasing oocyte meiotic arrest. They are likely to have effects on the full maturation of the egg in preparation for embryonic development. The actions of the multiple lipid messengers generated downstream of mPRβ activation are likely to act through GPCRs and could signal through Gas or other Ga or even through Gβγ. Minor point 7 refers to the size of Western blots.

(11) Page 9, bottom: "...one would predict activation of sphingosine kinases...." Couldn't it just be the activity of some constitutively active sphingosine kinase? Maybe replace "activation" with "activity".

A constitutively sphingosine kinase activity would not make sense as it needs to be activated by P4.

(12) Sometimes the authors refer to concentrations in molar units plus a power of 10 (e.g. 10-5 M) and sometime in µM or nM, sometimes even within the same paragraph. This makes it unnecessarily difficult to compare. Please keep consistent.

We replaced all the concentrations through the text to M with scientific notation for consistency as requested by the Reviewer.

(13) Fig 3I: "Sphingosine kinase" is misspelled.

This has been corrected. We thank the Reviewer for catching it.

(14) Legend to Fig. 5: Please change "after P4 treatment in reticulocytes" to "after P4 treatment in reticulocyte lysates".

Done

(15) Fig 6J. Doesn't the MAPK cascade inhibit MYT1? I.e. shouldn't the arrow be -| rather than ->?

Yes the Reviewer is correct. This has been changed. We thank the Reviewer for noticing this error.

(16) Materials and Methods, second paragraph. Please change "inhibitor's studies" to "inhibitor studies".

Corrected thanks.

(17) Table 1: Please be consistent in how you write Cox-2.

Done.

**Reviewer #3:**
The findings are of potential broad interest, but I have some concerns with the pharmacological studies presented. Many of these inhibitors are used at high (double-digit micromolar) concentrations that could result in non-specific pharmacological effects and the authors have provided very little data in support of target engagement and selectivity under the multiple experimental paradigms. Importantly, several claims regarding lipid metabolism signaling in the context of oocyte maturation are made without critical validation that the intended target is inactivated with reasonable selectivity across the proteome. Several of the inhibitors used for pharmacology and metabolomics are known covalent inhibitors (JZL184 and MJN110) that can readily bind additional lipases depending on the treatment time and concentration.I did not find any data using the reported ABHD2 inhibitor (compound 183; PMID: 31525885). Is there a reason not to include this compound to complement the knockdown studies? I believe this is an important control given that not all lipid effects were reversed with ABHD2 knockdown. The proper target engagement and selectivity studies should be performed with this ABHD2 inhibitor.

We obtained aliquots the reported ABHD2 inhibitor compound 183 from Dr. Van Der Stelt and tested its effect on oocyte maturation at 10^-4^M using both low (10^-7^M) or high (10^-5^M) P4 concentration. Compound 183 partially inhibited P4-mediated oocyte maturation. The new data was added to the manuscript as Supplemental Figure 3D.

Additional comments:(1) Pristimerin was tested at low P4 concentration for effects on oocyte maturation. Authors should also test JZL184 and MJN110 under this experimental paradigm.

We have tested the effect of high concentration (2.10-^-5^M) of JZL184 or MJN110 on oocyte maturation at low P4 concentration (Author response image 3). MJN 110 did not have a prominent effect on oocyte maturation at low P4, whereas JZL184 inhibited maturation by 50%. However, this inhibition of maturation required concentrations of JZL 184 that are 10 times higher than those reported in rat and human cells (Cui et al., 2016; Smith et al., 2015), arguing against an important role for a monoacylglycerol enzymatic activity in inducing oocyte maturation.

**Author response image 3. sa4fig3:** The effect of MJN110 and JZL184 compounds on oocyte maturation at low P4 concentration. Oocytes were pre-treated for 2 hours with the vehicle or with the highest concentration of 2.10-^-5^ M for both JZL184 or MJN110, followed by overnight treatment with P4 at 10-^7^M. Oocyte maturation was measured as % GVBD normalized to control oocytes (treated with vehicle) (mean + SEM; n = 2 independent female frogs for each compound).

1. Figure 4A showed different ct values of ODC between Oocytes and spleen, please explain them in the text. There is not any description regarding spleen information in Figure 4A, please make it clear in the text.

We thank the Reviewer for this recommendation. The text was revised accordingly.

(3) For Figures 3A, E, and I, there are different concentration settings for comparing the activity, is it possible to get the curves based on the same set of concentrations? The concentration gradient didn't include higher concentration points in these figures, thus the related values are incorrect. Please set more concentration points to improve the figures. And for the error bar, there are different display formats like Figure 4c and 4d, etc. Please uniform the format for all the figures. Additionally, for the ctrl. or veh., please add an error bar for all figures.

Some of the drugs tested were toxic to oocytes at high concentrations so the dose response was adjusted accordingly. The graphs were plotted to encompass the entire tested dose response. We could have plotted the data on the same x-axis range but that would make the figures uneven and awkward.

We are not clear what the Reviewer means by “The concentration gradient didn't include higher concentration points in these figures, thus the related values are incorrect.”

The error bars for all dose responses are consistent throughout all the Figures. They are different from those on bar graphs to improve clarity. If the Reviewer wishes to have the error bars on the bar graphs and dose response the same, we are happy to do so.

For the inhibitor studies the data were normalized on a per frog basis to control for variability in the maturation rate in response to P4, which varies from frog to frog. It is thus not possible to add error bars for the controls.

(4) Please check the sentence "However, the concentration of HA130...... higher that......'; Change "IC50" to "IC50" in the text and tables. Table 1 lists IC50 values in the literature, but the references are not cited. Please include the references properly. For the IC50 value obtained in the research, please include the standard deviation in the table. For reference parts, Ref 1, 27, 32, 46, doublecheck the title format.

We edited the sentence as follows to be more clear: “However, this inhibition of maturation required high concentrations of HA130 -at least 3 orders of magnitude higher that the reported HA130 IC_50_-…”

We changed IC50 to subscript in Table 1.

We added the relevant references in Table 1 to provide context for the cited IC50 values for the different inhibitors used.

We added SEM to the IC_50_ for inhibition of oocyte maturation values in Table 1.

We checked the titles on the mentioned references and cannot identify any problems.

References

Cui, Y., Prokin, I., Xu, H., Delord, B., Genet, S., Venance, L., and Berry, H. (2016). Endocannabinoid dynamics gate spike-timing dependent depression and potentiation. eLife *5*, e13185.

Nader, N., Dib, M., Hodeify, R., Courjaret, R., Elmi, A., Hammad, A.S., Dey, R., Huang, X.Y., and Machaca, K. (2020). Membrane progesterone receptor induces meiosis in *Xenopus* oocytes through endocytosis into signaling endosomes and interaction with APPL1 and Akt2. PLoS Biol *18*, e3000901.

Smith, M., Wilson, R., O'Brien, S., Tufarelli, C., Anderson, S.I., and O'Sullivan, S.E. (2015). The Effects of the Endocannabinoids Anandamide and 2-Arachidonoylglycerol on Human Osteoblast Proliferation and Differentiation. PloS one *10*, e0136546.